# PRINCIPAL-AGENT REINFORCEMENT LEARNING: ORCHESTRATING AI AGENTS WITH CONTRACTS

## ABSTRACT

The increasing deployment of AI is shaping the future landscape of the internet, which is set to become an integrated ecosystem of AI agents. Orchestrating the interaction among AI agents necessitates decentralized, self-sustaining mechanisms that harmonize the tension between individual interests and social welfare. In this paper we tackle this challenge by synergizing reinforcement learning with principal-agent theory from economics. Taken separately, the former allows unrealistic freedom of intervention, while the latter struggles to scale in sequential settings. Combining them achieves the best of both worlds. We propose a framework where a principal guides an agent in a Markov Decision Process (MDP) using a series of contracts, which specify payments by the principal based on observable outcomes of the agent's actions. We present and analyze a meta-algorithm that iteratively optimizes the policies of the principal and agent, showing its equivalence to a contraction operator on the principal's Q-function, and its convergence to subgame-perfect equilibrium. We then scale our algorithm with deep Q-learning and analyze its convergence in the presence of approximation error, both theoretically and through experiments with randomly generated binary game-trees. Extending our framework to multiple agents, we apply our methodology to the combinatorial Coin Game. Addressing this multi-agent sequential social dilemma is a promising first step toward scaling our approach to more complex, real-world instances.

## 1 INTRODUCTION

The deployment of AI agents is becoming increasingly prevalent across every facet of society (see, e.g., the survey of Wang et al., 2024a). Recent advancements have enabled large language models to leverage external tools via function-calling capabilities (e.g. Srinivasan et al., 2023; Li et al., 2024). For example, models like GPT-4 can now interface with databases, APIs, and other AI agents, significantly broadening their range of applications. Looking ahead, we anticipate the future internet to connect not only traditional computing devices but also everyday objects powered by AI agents. These developments mark a shift toward more integrated agent ecosystems, and necessitate *scalable* mechanisms for orchestrating efficient interactions among AI agents in a decentralized and heterogeneous environment.

However, learning efficient interactions is inherently complex. AI agents are typically hosted by different entities, and the parties involved in these interactions have individual interests, leading to a natural tension between their incentives and long-term overall welfare. Sequential social dilemmas (SSDs) (Leibo et al., 2017) capture this tension, highlighting how optimization of each agent's own reward in a Markovian environment may result in suboptimal global outcomes. Multi-agent reinforcement learning (MARL) attempts to address this challenge using methods like *reward shaping*, which adjusts the agents' reward functions to align their incentives. While effective at optimizing global objectives, these methods typically assume centralized control over each agent's training procedure, providing unrealistic freedom of intervention to their designer, as well as overlooking the costs of said interventions.

This discrepancy prompts us to look to the economics literature, where complex incentive structures between independent self-interested parties are focal. One of the primary directions in this line of work is the celebrated *principal-agent theory* (e.g., Holmström (1979); Grossman & Hart (1983); Rogerson (1985); Spear & Srivastava (1987)). It distinguishes between agents, who directly take

costly actions to complete tasks, and the principal, who receives rewards from the execution of those tasks. The main tool of this theory are *contracts*, which define payments from the principal to the agent, based on observable outcomes of the agents' actions. Conceptually, this theory fits well with the requirements of modeling interaction among AI agents: Some applications may explicitly involve a principal – for example, a navigation app benefiting from drivers exploring new routes (Ben-Porat et al., 2024). In other applications, including general frameworks like SSDs, a principal can serve as a helpful abstraction that represents social objectives and long-term global welfare.

Contract theory was recognized with the Nobel Prize in Economics in 2016, and has recently garnered interest from the computer science community (e.g., Ho et al. (2016); Dütting et al. (2019); Dütting et al. (2023)). However, the learning of contracts has largely been confined to single-round settings or relies on strict assumptions about agent capabilities (Zhu et al., 2023a; Wang et al., 2023; Guruganesh et al., 2024; Scheid et al., 2024). These limitations leave contract theory inadequately equipped to handle the complexities of future AI agent ecosystems. Motivated by this gap, we aim to synergize reinforcement learning and contract theory—we propose learning contracts as a scalable mechanism to align agents' incentives, and employ (multi-agent) reinforcement learning to extend contract learning to general, sequential settings.

**Our Contribution**. We begin by formulating a general framework for *principal-agent reinforcement learning* with a single agent (Section 2). In this framework, the agent learns a policy for an MDP, and the principal learns to guide the agent using a series of contracts. Importantly, we explicitly model the (usually misaligned) preferences of the principal and the agent.

We first examine this problem from a purely economic perspective (Section 3) with full access to the MDP, thus alleviating the need for learning. We focus on the standard solution concept of *subgame-perfect equilibrium* (SPE, Section 3.1). We formulate a meta-algorithm (Algorithm 1) that iteratively optimizes the principal's and agent's policies in their respective MDPs with the other player's policy being fixed. We show that the meta-algorithm finds an SPE in a finite-horizon game in at most $T + 1$ iterations, where $T$ is the horizon (Theorem 3.3). We also show that the meta-algorithm can be viewed as iteratively applying a contraction operator to the principal's Q-function (Theorem 3.4), ensuring its monotonic improvement with each iteration.

Next, in Section 4, we turn to the standard model-free RL setting where the MDP is unknown, and the policies are learned by sampling stochastic transitions and rewards through interacting with the MDP. We instantiate the meta-algorithm (see Algorithm 2) by solving both principal's and agent's MDPs with (deep) Q-learning and apply it in a two-phase setup. First, we train the principal assuming centralized access to the agent's optimization problem (of maximizing its Q-function estimate given a contract in a state). Then, we lift the assumption and validate the learned principal's policy against a black-box oracle agent, mimicking its execution in the real world. We analyze this learning setup theoretically in the presence of approximation errors, including a tool for addressing potential deviations of the oracle agent through extra payments, which we call *nudging*. We present a host of experiments in a controlled environment with randomly generated binary game-trees, and verify that our algorithm approximates SPE well despite approximation errors and even without nudging.

Finally, in Section 5, we extend our model to multiple agents and identify *sequential social dilemmas* (SSDs, Leibo et al. (2017)) as a natural testbed for principal-agent RL. Complementing the prior work that has approached SSDs through multi-agent RL and reward shaping, we adopt a more economic perspective by framing interventions into reward functions as payments made by an external, benevolent principal. The principal's objective is to maximize social welfare through *minimal* intervention. We present extensive experiments with a prominent SSD, the Coin Game (Foerster et al., 2018), where two agents (blue and a red) collect blue and red coins in a grid world. Collecting a coin of the matching color is five times as rewarding to an agent as collecting one of the opposite color, suggesting cooperation by collecting only coins of agents' respective colors. This seemingly innocent grid world is highly combinatorial in terms of states (all possible agent and coin positions on a 7x7 grid), actions (all possible sequences across 50 timesteps), and agents (all possible opponent policies). The complexity is further elevated when adding the principal to the mix. Through our two-phase experimental setup – centralized training followed by black-box validation – we verify the approximate convergence of our method to SPE in the Coin Game. We also compare to a simpler, hand-coded payment scheme inspired by Christoffersen et al. (2023), and observe that with the same amount of subsidy, this less-targeted intervention achieves a significantly lower welfare level.

## 1.1 RELATED WORK

Our work fits into the wider literature on automated mechanism design (Conitzer & Sandholm, 2002), in particular approaches based on deep learning (Dütting et al., 2019; Wang et al., 2023) also known as differentiable economics (Dütting et al., 2024). Most closely related from this line of work, Wang et al. (2023) consider stateless one-shot contracting problems and provide a network architecture for capturing the discontinuities in the principal's utility function. We differ from this work in our focus on sequential contracting problems and the entailing unique challenges.

There is a number of algorithmic works that focus on repeated principal-agent interactions on MDPs, including work on *environment design* and *policy teaching* (Zhang & Parkes, 2008; Zhang et al., 2009b; Yu & Ho, 2022; Ben-Porat et al., 2024). Our approach differs from these earlier works in several ways, including that we actively search for the best (unconstrained) equilibrium in the game between the principal and the agent through reinforcement learning. A closely related line of work, including Gerstgrasser & Parkes (2023), is concerned with learning Stackelberg equilibria in general leader-follower games, including games on MDPs. Our work differs in its focus on SPE, which is the more standard equilibrium concept in dynamic contract design problems. Several works have studied repeated contract design problems from a no-regret online-learning perspective (Ho et al., 2016; Zhu et al., 2023a; Guruganesh et al., 2024; Scheid et al., 2024). However, these works are typically limited to stateless and/or non-sequential interactions. A prominent exception is a contemporaneous study by Wu et al. (2024) that introduces a model of principal-agent MDPs nearly identical to ours, barring an important notational distinction of encoding outcomes as next states.[1] However, their and our studies pursue orthogonal algorithmic developments: whereas they treat contract policies as arms of a bandit and minimize regret, we rely on deep RL to scale to large MDPs and multiple agents.

Starting with the work of Leibo et al. (2017), there is a huge literature on SSDs. Most closely related in this direction is the work by Christoffersen et al. (2023) on multi-agent RL and applications to SSDs. This work pursues an approach in which one of the agents (rather than the principal) proposes a contract (an outcome-contingent, zero-sum reward redistribution scheme), and the other agents can either accept or veto. They consider several SSDs and show how hand-crafted contract spaces strike a balance between generality and tractability, and can be an effective tool in mitigating social dilemmas. An important distinction of our work is that we distinguish between principal and agent(s), and insist on the standard limited liability requirement from economics. Furthermore, in our approach the principal *learns* the conditions for payments, allowing it to utilize contracts in their full generality. Also, our method has an interpretation as learning $k$-implementation (Monderer & Tennenholtz, 2003). We provide additional details on the related literature in Appendix A.

## 2 PROBLEM SETUP

In this section, we first introduce the classic (limited liability) contract design model of Holmström (1979) and Grossman & Hart (1983) (see Section 2.1), and then propose its extension to MDPs (Section 2.2). We defer further extension to multi-agent MDPs to Section 5.1.

### 2.1 THE STATIC HIDDEN-ACTION PRINCIPAL-AGENT PROBLEM

In a classic principal-agent problem, the principal wants the agent to invest effort in a task. The agent has a choice between several *actions* $a \in A$ with different *costs* $c(a)$, interpreted as effort levels. Each action stochastically maps to *outcomes* $o \in O$ according to a distribution $\mathcal{O}(a)$, with higher effort levels more likely to result in rewarding outcomes, as measured by the principal's *reward* function $r^p(o)$. By default, a rational agent would choose the cost-minimizing action. To incentivize the agent to invest effort, the principal offers a *contract* $b$ prior to the action choice. Crucially, the principal is unable to directly view the agent's hidden action, so the contractual payments $b(o)$ are defined per outcome $o \in O$. Payments are always non-negative – this is referred to as *limited liability*. The principal seeks an *optimal* contract: a payment scheme $b$ that maximizes the principal's $\mathbb{E}[utility]$, $\mathbb{E}_{o \sim \mathcal{O}(a)}[r^p(o) - b(o)]$, given that the agent best-responds with $a = \arg\max_{a'} \mathbb{E}_{o \sim \mathcal{O}(a')}[b(o)] - c(a')$.

If costs and outcome distributions are known, the optimal contract can be precisely found using Linear Programming (LP): for each action $a \in A$, find the contract that implements it (makes the

---

[1]Our model is technically more general by allowing stochasticity in both the outcome and transition functions.

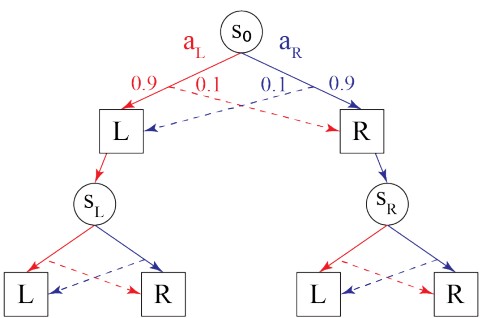

Figure 1: Example of a principal-agent MDP with three states $S = \{s_0, s_L, s_R\}$. In each state, the agent can take one of two actions: noisy-left $a_L$, which is costly and leads to outcomes $L$ and $R$ with probabilities 0.9 and 0.1, and noisy-right $a_R$, which is free and has the likelihood of $L$ and $R$ reversed. The principal's rewards in any state $s \in S$ for outcomes $L, R$ are $r^p(s, L) = \frac{14}{9}, r^p(s, R) = 0$, while those of the agent for the actions are $r(s, a_L) = -\frac{4}{5}, r(s, a_R) = 0$. For analysis, see Appendix B.1.

agent prefer it to other actions) through a minimal expected payment, and then choose the best action to implement. Otherwise or if the LPs are infeasible, an approximation can be obtained by training a neural network on a sample of past interactions with the agent (Wang et al., 2023).

## 2.2 HIDDEN-ACTION PRINCIPAL-AGENT MDPS

In order to extend contract design to MDPs, we assume a principal-agent problem in each state of the MDP, and let the outcomes additionally define its (stochastic) transitioning to the next state. Formally, a *hidden-action principal-agent MDP* is a tuple $\mathcal{M} = (S, s_0, A, B, O, \mathcal{O}, \mathcal{R}, \mathcal{R}^p, \mathcal{T}, \gamma)$. As usual in MDPs, $S$ is a set of states, $s_0 \in S$ is the initial state, and $A$ is a set of $n$ agent's actions $a \in A$. Additionally, $O$ is a set of $m$ outcomes $o \in O$, $B \subset \mathbb{R}_{\geq 0}^m$ is a set of principal's actions (contracts) $b \in B$, and $b(o)$ is the $o$-th coordinate of a contract (payment). In each state, the outcome is sampled based on the agent's action from a distribution $\mathcal{O}(s, a)$, where $\mathcal{O} : S \times A \to \Delta(O)$ is the outcome function. Then, the agent's reward is determined by the reward function $\mathcal{R}(s, a, b, o) = r(s, a) + b(o)$ for some $r(s, a)$. Likewise, the principal's reward is determined by $\mathcal{R}^p(s, b, o) = r^p(s, o) - b(o)$ for some $r^p(s, o)$; note that the agent's action is hidden and thus $\mathcal{R}^p$ does not explicitly depend on it. Based on the outcome, the MDP transitions to the next state $s' \sim \mathcal{T}(s, o)$, where $\mathcal{T} : S \times O \to \Delta(S)$ is the transition function. Finally, $\gamma \in [0, 1]$ is the discount factor. For an example of a principal-agent MDP, see Figure 1. In the main text, we focus on MDPs with a finite time horizon $T$. We assume w.l.o.g. that each state is uniquely associated with a time step (see Appendix B.4).

We analyze the principal-agent MDP as a *stochastic game* $\mathcal{G}$ (can be seen as *extensive-form* when the horizon is finite), where two players maximize their long-term payoffs. The game progresses as follows. At each timestep $t$, the principal observes the state $s_t$ of the MDP and constructs a contract $b_t \in B$ according to its *policy* $\rho : S \to \Delta(B)$. Then, the agent observes the pair $(s_t, b_t)$ and chooses an action $a_t \in A$ according to its policy $\pi : S \times B \to \Delta(A)$. After this, the MDP transitions, and the interaction repeats. Both players maximize their *value* functions: the principal's value function, $V^\rho(\pi)$, of a policy $\rho$ given the agent's policy $\pi$ is defined in a state $s$ by $V^\rho(s \mid \pi) = \mathbb{E}[\sum_t \gamma^t \mathcal{R}^p(s_t, b_t, o_t) \mid s_0 = s]$; likewise, the agent's value function $V^\pi(\rho)$ is defined in a state $s$ and given a contract $b$ by $V^\pi(s, b \mid \rho) = \mathbb{E}[\sum_t \gamma^t \mathcal{R}(s_t, a_t, b_t, o_t) \mid s_0 = s, b_0 = b]$. Players' *utilities* in the game are their values in the initial state. Additionally, define the players' *Q-value* functions $Q^\rho(\pi)$ and $Q^\pi(\rho)$ by $Q^\rho(s, b \mid \pi) = V^\rho(s \mid b_0 = b, \pi)$ and $Q^\pi((s, b), a \mid \rho) = V^\pi(s, b \mid a_0 = a, \rho)$.

A special case of the principal-agent MDP trivializes hidden actions by making the outcome function deterministic and bijective – essentially the outcome reveals the action to the principal; the resulting *observed-action* model is similar to Ben-Porat et al. (2024). For an explicit comparison of the two models with each other and with standard MDPs, see Appendix B.0.

---

**Algorithm 1** Meta-algorithm for finding SPE

---
1: Initialize the principal's policy $\rho$ arbitrarily      $\triangleright$ e.g., $\forall s \in S : \rho(s) = \mathbf{0}$
2: **while** $\rho$ not converged **do**      $\triangleright$ Inner-outer optimization loop
3:      Solve the agent's MDP: find $\pi := \pi^*(\rho)$      $\triangleright$ Inner optimization level
4:      Solve the principal's MDP: find $\rho := \rho^*(\pi)$      $\triangleright$ Outer optimization level
5: **return** $(\rho, \pi)$

---

## 3   Purely Economic Setting

In this section, we define our solution concept for principal-agent stochastic games (Section 3.1) and introduce a meta-algorithm that finds this solution (Section 3.2). We assume full access to the MDP model (including transition and reward functions) and defer the learning setting to the next section. We focus on finite-horizon MDPs; we analyze infinite horizon in Appendix B.

### 3.1   Subgame-Perfect Equilibrium (SPE)

In what follows let $\mathcal{G}$ be a principal-agent stochastic game. Let a *subgame* of $\mathcal{G}$ in state $s \in S$ be a game $\mathcal{G}'$ defined by replacing $s_0$ with $s$, and $S$ with a subset of states that can be reached from $s$.

**Observation 3.1.** Fixing one player's policy in $\mathcal{G}$ defines a (standard) MDP for another player. In particular, a principal's policy defines the *agent's MDP* by modifying the reward function through contracts; likewise, an agent's policy defines the *principal's MDP* by modifying the transition and reward functions through the agent's responses to contracts. For exact formulations of these MDPs, see Appendix B.2. The optimal policies in both MDPs can be assumed w.l.o.g. to be deterministic (in single-agent setting), and can be found with any suitable dynamic programming or RL algorithm (Sutton & Barto, 2018).

**Agent's perspective.** In the agent's MDP defined by $\rho$, refer to the optimal policy as *best-responding* to $\rho$ (where ties are broken in favor of the principal, as is standard in contract design). Define a function $\pi^*$ that maps a principal's policy $\rho$ to the best-responding policy $\pi^*(\rho)$. Denote the action prescribed by $\pi^*(\rho)$ in state $s$ given contract $b$ by $\pi^*(s, b \mid \rho) \equiv \pi^*(\rho)(s, b)$. Note that the best-responding policy is defined in $s$ for any $b \in B$ and is not limited to the principal's action $\rho(s)$.

**Principal's perspective.** Similarly, in the principal's MDP defined by $\pi$, refer to the optimal policy as *subgame-perfect* against $\pi$. Define a function $\rho^*$ that maps an agent's policy $\pi$ to the subgame-perfect policy $\rho^*(\pi)$. Denote the contract prescribed by $\rho^*(\pi)$ in state $s$ by $\rho^*(s \mid \pi) \equiv \rho^*(\pi)(s)$. In all states, this policy satisfies: $\rho^*(s \mid \pi) \in \arg\max_b Q^*(s, b \mid \pi)$, where $Q^*(s, b \mid \pi) \equiv Q^{\rho^*(\pi)}(s, b \mid \pi)$ – that is, in each subgame, the principal takes the optimal action.

**Definition 3.2.** A *subgame-perfect equilibrium (SPE)* of $\mathcal{G}$ is a pair of policies $(\rho, \pi)$, where the principal's policy $\rho \equiv \rho^*(\pi)$ is subgame-perfect against an agent that best-responds with $\pi \equiv \pi^*(\rho)$.

SPE is the standard solution concept for extensive-form games (Osborne, 2004). It always exists and is essentially unique (see Lemma B.12 for completeness). Compared to non-subgame-perfect solutions like the well–studied *Stackelberg equilibrium*, SPE can lose utility for the principal. However, it is considered more realistic since it disallows *non-credible threats*, i.e., commitments by the principal to play a suboptimal contract if one of the subgames is reached. Moreover, Gerstgrasser & Parkes (2023) show that learning a Stackelberg equilibrium necessitates the principal to go through long episodes of observing the learning dynamic of the agent, with only sparse rewards (see also Brero et al. (2022)). SPE, in contrast, naturally fits RL, as both players' policies solve their respective MDPs. We demonstrate the difference on our running example in Appendix B.1.

### 3.2   Meta-Algorithm for Finding SPE

Algorithm 1 presents a general pipeline for finding SPE in a principal-agent stochastic game. It can be seen as an inner-outer (bilevel) optimization loop, with agent and principal optimization respectively constituting the inner and outer levels. We refer to this as a meta-algorithm, as we do not yet specify how to perform the optimization (in Lines 3 and 4). Superficially, this approach resembles the use of

Table 1: Correctness of the meta-algorithm in different scenarios

|  | finite horizon $T$ | infinite horizon |
|---|---|---|
| hidden action | finds SPE in $T + 1$ iterations (Theorem 3.3) | may diverge (Appendix B.6) |
| observed action | finds SPE that is also Stackelberg in 1 iteration (Appendix B.7) | |

bilevel optimization for learning optimal reward shaping (Stadie et al., 2020; Hu et al., 2020; Chen et al., 2022; Wang et al., 2022; Chakraborty et al., 2023; Lu, 2023). The crucial difference of that setting is that the two levels optimize the same downstream task rather than distinct and possibly conflicting objectives of principal and agent.

It is well-known that SPE of an extensive-form game can be found with backward induction (e.g., see Section 5.6 of Osborne (2004)). Theorem 3.3 states that the proposed meta-algorithm also finds SPE. The proof, provided in Appendix B.4, essentially shows that it performs backward induction implicitly: each iteration of the meta-algorithm, the players' policies reach SPE in an expanding set of subgames, starting from terminal states and ending with the initial state. The proof does not rely on the specifics of our model and applies to any game where players move sequentially, and the agent observes and can respond to *any* principal's action in a state.

**Theorem 3.3.** *Given a principal-agent stochastic game $\mathcal{G}$ with a finite horizon $T$, the meta-algorithm finds SPE in at most $T + 1$ iterations.*

The meta-algorithm has several unique advantages. First, as we discuss in Section 4, both inner and outer optimization tasks can be instantiated with Q-learning. This removes the need to know the model of the MDP and allows handling of large-scale MDPs by utilizing deep learning. Second, it can also be seen as iteratively applying a *contraction* operator, which we formulate as a theorem:

**Theorem 3.4.** *Given a principal-agent finite-horizon stochastic game $\mathcal{G}$, each iteration of the meta-algorithm applies to the principal's Q-function an operator that is a contraction in the sup-norm.*

The proof is provided in Appendix B.5. This property implies that each iteration of the meta-algorithm monotonically improves the principal's policy in terms of its Q-function converging. This has a practical advantage: if meta-algorithm is terminated early, the policies still partially converge to SPE.

As an independent observation, Theorem 3.3 complements the theoretical results of Gerstgrasser & Parkes (2023). Specifically, their Theorem 2 presents an example where RL fails to converge to a *Stackelberg* equilibrium if the agent 'immediately' best responds. This procedure is our Algorithm 1, with agent optimization solved by an oracle, and principal optimization performed with RL. Our Theorem 3.3 complements their negative result by showing that such a procedure converges to *SPE*.

Finally, while we focus on finite-horizon hidden-action MDPs in the main text, we also analyze the other scenarios in the Appendix. Our findings are summarized in Table 1.

## 4 LEARNING SETTING

In this section, we develop an RL approach to principal-agent MDPs by solving both inner and outer optimization tasks of the meta-algorithm with Q-learning. These tasks correspond to finding optimal policies in the respective agent's and principal's MDPs defined in Observation 3.1. We operate in a standard model-free RL setting, where learning is performed through interactions with a black-box MDP. For both principal and agent, we introduce modified Q-functions, which we formally derive as fixed points of contraction operators in Appendix B.3. We detail deep implementations in Appendix D.

**Agent's learning.** Consider the agent's MDP defined by principal's policy $\rho$. The best-responding agent's Q-function in a state $s$, $Q^*((s, b), a \mid \rho)$, depends on the observed contract $b$ – particularly on the expected payment. This effect can be isolated by applying the Bellman optimality operator:

$$Q^*((s, b), a \mid \rho) = \mathbb{E}_{o \sim \mathcal{O}(s,a)}[b(o)] + \overline{Q}^*(s, a \mid \rho), \tag{1}$$

where $\overline{Q}^*(s, a \mid \rho) = [r(s,a) + \gamma\mathbb{E}\max_{a'} Q^*((s', \rho(s')), a' \mid \rho)]$ is the *truncated* optimal Q-function, which represents the agent's expected long-term utility barring the immediate payment. Our approach to training the agent (solving inner optimization) is to learn the truncated Q-function and compute the Q-function through Equation (1). This way, the Q-function is defined for any $b \in B$ in $s$, under an assumption that the principal plays according to $\rho$ in future (e.g., $\rho(s')$ in the next state). From the agent's perspective, this is justified in SPE, where $\rho$ is optimal for the principal in all future subgames. Note that computing the expected payment requires the outcome distribution – if unknown, it can be approximated as a probabilistic classifier (more on this in Appendix D.1).

**Principal's learning.** Consider the principal's MDP defined by a best-responding agent $\pi^*(\rho)$ for an arbitrary $\rho$. The basic idea is to divide the principal's learning problem into two parts: 1) learn the agent's policy that the principal wants to implement (*recommends* to the agent), and 2) compute the optimal contracts that implement it (the *minimal implementation*) using Linear Programming (LP). Essentially, this extends the classic LP approach from static contract design described in Section 2.1.

To approach the first subproblem, we need an analogue of the principal's Q-function that is a function of an agent's action. To this end, we define the *contractual* Q-function $q^*(\pi^*(\rho)) : S \times A \to \mathbb{R}$ by

$$q^*(s, a^p \mid \pi^*(\rho)) = \max_{\{b \mid \pi^*(s, b \mid \rho) = a^p\}} Q^*(s, b \mid \pi^*(\rho)), \tag{2}$$

which can be interpreted in $s$ as the maximal principal's Q-value that can be achieved by implementing $a^p \in A$. To compute the optimal contract $\arg\max_b Q^*(s, b \mid \rho)$ using $q^*(\pi^*(\rho))$, we can select the optimal action to implement as $\arg\max_{a^p} q^*(s, a^p \mid \pi^*(\rho))$, and then find the corresponding contract as a solution to the conditional maximization in Equation (2). This conditional maximization is the second subproblem defined above. We solve it as an LP (for details, see Appendix B.8):

$$\max_{b \in B} \quad \mathbb{E}_{o \sim \mathcal{O}(s, a^p)}[-b(o)] \quad \text{s.t.}$$
$$\mathbb{E}_{o \sim \mathcal{O}(s, a^p)}[b(o)] + \overline{Q}^*(s, a^p \mid \rho) \geq \mathbb{E}_{o \sim \mathcal{O}(s, a)}[b(o)] + \overline{Q}^*(s, a \mid \rho) \quad \forall a \in A. \tag{3}$$

Solving this LP requires access to the agent's truncated Q-function $\overline{Q}^*$. Although this requirement is in line with the training phase of our setup, it can be alleviated, e.g., by approximating optimal contracts with deep learning (Wang et al., 2023). We do not explore this direction, preferring to focus here on the distinct learning problems originating from our MDP formulation.

**Approximation error.** In the idealized setting of Section 3, the inner and outer optimization tasks are solved exactly. However, learning is usually performed with approximation errors (due to early termination, function approximation, etc.). An issue central to the principal-agent theory is that, due to the principal's utility being discontinuous, even a slight misestimation of the optimal contract may have a devastating effect by changing the agent's response (see, e.g., Wang et al. (2023)). We study the effect of approximation error in our model through the following two-phase setup:

1. **Training:** The principal's policy is found with the meta-algorithm, instantiated with Q-learning (as described above) in the presence of approximation error.
2. **Validation:** The learned principal's policy is validated against a black-box oracle agent that exactly best-responds (without approximation error).

We analyze this setup theoretically in Appendix B.9 and summarize our findings in the following proposition:

**Proposition 4.1.** *At the training phase, if at each iteration $t$, the learning of the principal's (resp., agent's) Q-function is performed with an error of up to $\delta_t$ (resp., $\epsilon_t$), then the total error in $s_0$ is bounded by $D_0 = \sum_{t \in [0,T]} \gamma^t \delta_t$ (resp., $E_0 = \sum_{t \in [0,T]} \gamma^t \epsilon_t$). At the validation phase, the principal can be guaranteed a utility within $-2D_0 - 2\sum_t \gamma^t E_t / d_{\min}$ from the optimal, where $d_{\min}$ is a measure of how much the outcome distributions for different actions intersect (defined in (23)). This utility can be achieved through small additional payments that counteract the agent's deviations, which we call nudging.*

Additionally, we analyze the effect of approximation errors empirically. In Appendix D.1, we report the results of applying our algorithm to principal-agent MDPs represented by randomly sampled binary game-trees. We find that even without nudging, the learned principal's policy achieves a utility at validation within 1-2% from the optimal. Interestingly, this is achieved by choosing the optimal

action $a^p$ to recommend in only about 90% of all states, suggesting that errors likely occur in rarely visited or less impactful states. In the next section, we will continue our empirical investigation in a much more complex environment.

## 5 EXTENSION TO MULTI-AGENT RL

In this section, we explore an extension to multiple agents. We state the formal model in Section 5.1. We introduce *sequential social dilemmas* (SSDs) and the Coin Game in Section 5.2. We present experimental results in Section 5.3.

### 5.1 PROBLEM SETUP

Both our principal-agent model and our theory for the meta-algorithm can be extended to multi-agent MDPs. First, we formulate an analogous principal-multi-agent MDP, where a principal offers a contract to each agent, and payments are determined by the joint action of all agents. We treat joint actions as outcomes and omit hidden actions. Then, the theory extends by viewing all agents as a centralized super-agent that selects an equilibrium joint policy (in the multi-agent MDP defined by the principal). Finally, we address the issue of multiple equilibria by imposing an additional constraint on the incentive-compatibility of contracts, making our approach more robust to deviations. See also Appendix C, where we illustrate the multi-agent model on Prisoner's Dilemma.

A *principal-multi-agent MDP* is a tuple $\mathcal{M}_N = (S, s_0, N, (A_i)_{i \in N}, B, \mathcal{T}, (\mathcal{R}_i)_{i \in N}, \mathcal{R}^p, \gamma)$. The notation is as before, with the introduction of a set of $k$ agents, $N$, and the corresponding changes: $A_i$ is the action set of agent $i \in N$ with $n_i$ elements; $\mathbf{A}_N$ is the joint action set with $m = \prod_i n_i$ elements, defined as a Cartesian product of sets $A_i$; $B \subset \mathbb{R}^m_{\geq 0}$ is a set of contracts the principal may offer to an agent; $b_i$ denotes a contract offered to agent $i$, and $b_i(\mathbf{a})$ denotes a payment to $i$ determined by joint action $\mathbf{a} \in \mathbf{A}_N$; $\mathcal{T} : S \times \mathbf{A}_N \to \Delta(S)$ is the transition function; $\mathcal{R}_i : S \times \mathbf{A}_N \times B \to \mathbb{R}$ is the reward function of agent $i$ defined by $\mathcal{R}_i(s, \mathbf{a}, b_i) = r_i(s, \mathbf{a}) + b_i(\mathbf{a})$ for some $r_i$; $\mathcal{R}^p : S \times \mathbf{A}_N \times B^k \to \mathbb{R}$ is the principal's reward function defined by $\mathcal{R}^p(s, \mathbf{a}, \mathbf{b}) = r^p(s, \mathbf{a}) - \sum_i b_i(\mathbf{a})$. In our application, the principal's objective is to maximize agents' social welfare through minimal payments, so we define its reward by $r^p(s, \mathbf{a}) = \frac{1}{\alpha} \sum_i r_i(s, \mathbf{a})$, where $0 < \alpha < 1$ is a hyperparameter that ensures that payment minimization is a secondary criterion and does not hurt social welfare (we use $\alpha = 0.1$). Additionally, $\rho : S \to B^k$ is the principal's policy, and $\pi_i : S \times B \to \Delta(A_i)$ is an agent's policy. Because $B$ grows exponentially with the number of agents $k$, in our implementation, the principal gives an action recommendation to each agent, and the payments are determined after agents act.

Analogously to Observation 3.1, a fixed principal's policy $\rho$ defines a multi-agent MDP by changing the agents' reward functions. Importantly, this MDP can itself be analyzed as a Markov game between the agents (Littman, 1994). In this game, we use a basic solution concept called Markov Perfect Equilibrium (MPE, Maskin & Tirole (2001)), defined as a tuple of agents' policies $\boldsymbol{\pi}^*(\rho)$ such that the following holds:

$$\forall s, b_i, i, \pi_i : V_i^{\pi_i^*(\rho)}(s, b_i \mid \boldsymbol{\pi}_{-i}^*(\rho), \rho) \geq V_i^{\pi_i}(s, b_i \mid \boldsymbol{\pi}_{-i}^*(\rho), \rho). \tag{4}$$

Here, $\boldsymbol{\pi}_{-i}^*(\rho)$ denotes equilibrium policies of agents other than $i$, and $V_i^{\pi_i}(\cdot \mid \boldsymbol{\pi}_{-i}, \rho)$ is the value function of agent $i$ playing $\pi_i$ given that other agents play $\boldsymbol{\pi}_{-i}$ and principal plays $\rho$. In MPE, no agent has a beneficial, unilateral deviation in any state.

Call MPE $\boldsymbol{\pi}^*(\rho)$ a *best-responding joint policy*; in case there are multiple, assume that agents break ties in favor of the principal. This assumption allows agents to freely coordinate the joint action, similarly to the equilibrium oracle of Gerstgrasser & Parkes (2023). The principal's *subgame-perfect policy* is defined by $\rho^*(s \mid \boldsymbol{\pi}) \in \arg\max_b Q^*(s, b \mid \boldsymbol{\pi})$, and an SPE is defined as a pair of policies $(\rho, \boldsymbol{\pi}^*(\rho))$ that are respectively subgame-perfect and best-responding against each other.

With this, our theory in Section 3 can be extended to the multi-agent model. Particularly, convergence proofs of Algorithm 1 apply with the swap of notation and the new definition of best-responding policy $\boldsymbol{\pi}^*(\rho)$. However, implementing this theory is problematic because of how strong the tie-breaking assumption is. In practice, there is no reason to assume that decentralized learning agents will converge to any specific equilibrium. For example, even in simple games such as Iterated Prisoner's Dilemma, RL agents typically fail to converge to cooperative Tit-for-Tat equilibrium (Foerster et al., 2018; Willis et al., 2023).

To provide additional robustness, we specify the principal's policy $\rho$ in SPE by requiring that it implements MPE $\boldsymbol{\pi}^*(\rho)$ *in dominant strategies*. Specifically, $\rho$ must additionally satisfy:

$$\forall s, b_i, i, \pi_i, \boldsymbol{\pi}_{-i}: \quad V_i^{\pi_i^*(\rho)}(s, b_i \mid \boldsymbol{\pi}_{-i}, \rho) \geq V_i^{\pi_i}(s, b_i \mid \boldsymbol{\pi}_{-i}, \rho). \tag{5}$$

This way, an agent prefers $\pi_i^*(\rho)$ regardless of other players' policies. We refer to contracts that make a strategy profile dominant as *Incentive-Compatible (IC)*, and to the IC contracts that minimize payments as a *minimal implementation*. In these terms, the principal's objective is to learn a social welfare maximizing strategy profile and its minimal implementation. This solution concept is inspired by the k-implementation of Monderer & Tennenholtz (2003).

## 5.2 A SEQUENTIAL SOCIAL DILEMMA: THE COIN GAME

In this section, we augment a multi-agent MDP known as the *Coin Game* (Foerster et al., 2018) with a principal and conduct a series of experiments. These experiments complement our theoretical results by empirically demonstrating the convergence of our algorithm to SPE in a complex multi-agent setting. In addition, we find a minimal implementation of a strategy profile that maximizes social welfare in a complex SSD, which is a novel result of independent interest.

**Environment.** The Coin Game is a standard benchmark in Multi-Agent RL that models an SSD with two self-interested players that, if trained independently, fail to engage in mutually beneficial cooperation. This environment is highly combinatorial and complex due to a large state space and the inherent non-stationarity of simultaneously acting and learning agents. Each player is assigned a color, red or blue, and collects coins that spawn randomly on a grid. Players earn $+1$ for collecting a coin of their color and $+0.2$ for other coins. [2] Our experiments are carried out on a $7 \times 7$ grid with each episode lasting for $50$ time steps; results on a smaller grid are provided in Appendix D.2.

**Experimental procedure.** We implement the two-phase approach described in Section 4. We parameterize principal's and agent's Q-functions as the respective deep Q-networks, $\theta$ and $\phi$. The principal is trained to maximize social welfare (sum of rewards) of the agents using VDN (Sunehag et al., 2018), and the agents' networks share parameters. Given the complexity of the Coin Game, validating the resulting policy against oracle agents is infeasible. Instead, at validation phase we train black-box agents from scratch as independent DQNs. For details and pseudocode, see Appendix D.2.

For baselines, we compare against a heuristic that distributes a constant proportion of social welfare. For a fair comparison, this proportion is set to be exactly equal to the proportion that our method ends up paying after the validation phase. This heuristic is at the core of approaches that improve social welfare through contractual agreements between agents (Christoffersen et al. (2023), see Appendix A). We also include a *selfish baseline* with self-interested agents in the absence of contracts, and an *optimal baseline* where agents are fully cooperative and directly maximize social welfare. These are instances of the constant proportion baseline with the proportion set to $0$ and $1$, respectively.

## 5.3 EXPERIMENTAL RESULTS

The results are presented in Figure 2. The social welfare metric (Fig. 2a) shows a gap between the performances of selfish and optimal baselines, confirming the presence of a conflict of interests. During training, our algorithm finds a joint policy that matches the optimal performance, and is implemented with an average payment of just above $30\%$ of social welfare, substantially reducing the intervention into the agents' rewards compared to the optimal baseline (Fig. 2b).

After the validation phase, the social welfare and the proportion paid to agents closely match the corresponding metrics in training. Furthermore, the agents follow the principal's recommendations in around $80\%$ to $90\%$ of states in an average episode (Fig. 2c). These results suggest that the principal closely approximated the SPE, as agents deviate only rarely and in states where it does not hurt social welfare. Given the challenges of convergence of independent RL agents to mutually beneficial equilibria, we find this success quite surprising, and attribute it to the IC property of the principal. From the perspective of an agent, there could be other optimal policies against different opponents, but following the principal's recommendations is robust against *any* opponent.

---

[2] We use the rllib code but remove the penalty a player incurs if its coin is picked up by the other player.

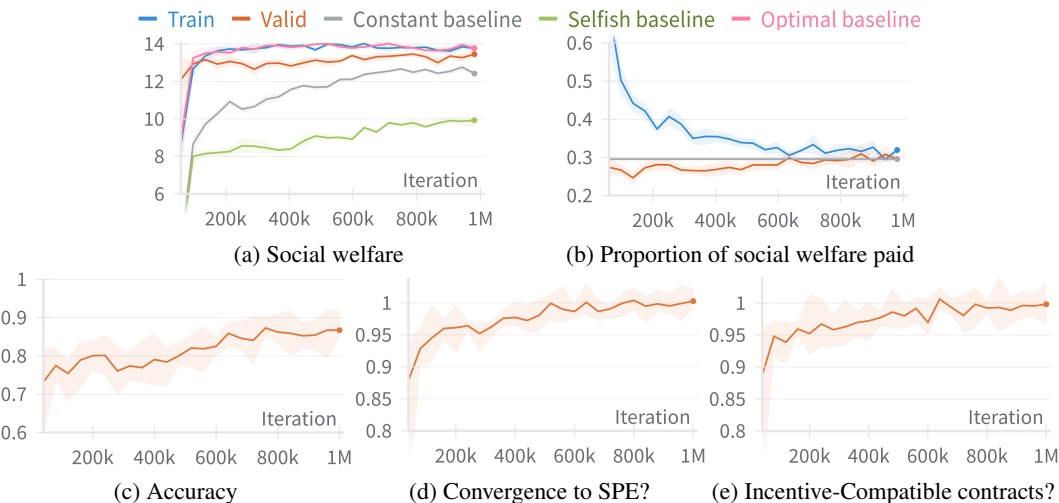

Figure 2: Learning curves in the Coin Game. See Section 5.3 for plot explanations. Shaded regions represent standard errors in the top plots and min-max ranges in the bottom plots.

We also see that the constant proportion baseline is much less effective than our algorithm when given the same amount of budget. The heuristic scheme overpays in some states while underpaying in others—incentivizing agents to selfishly deviate from a welfare-optimizing policy.

These results suggest the algorithm's convergence to SPE and the IC property of contracts. To further verify this, we collect additional metrics throughout the validation phase. Consider the perspective of the blue agent (Blue). At a given iteration, we fix the red agent's (Red's) policy, estimate Blue's utilities (average returns) under its policy and the recommended policy, and compare their ratio. In this scenario, if Red follows a recommended policy, then a utility ratio exceeding 1 would mean that there is a better policy for Blue than the recommended one, indicating a violation of the SPE condition (4) in $s_0$. We report this ratio in Figure 2d. Although agents occasionally discover slightly more profitable policies, the average utility ratio hovers around 1, indicating an approximate SPE. In the same scenario, if instead, Red acts according to its own policy, then a utility ratio exceeding 1 for Blue would indicate a violation of the IC conditions (5) in $s_0$. We report this ratio in Figure 2e. It behaves similarly, in that the average ratio again hovers around 1. We conclude that the principal is finding a good approximation to a minimal implementation that maximizes social welfare.

We report additional results for a smaller grid world in Appendix D.2, where we find that a form of nudging is necessary for good performance.

## 6 CONCLUSION

In this work, we develop a scalable framework that combines contract theory and reinforcement learning to address decentralized single- and multi-agent RL problems. Our framework offers formal definitions, a simple algorithmic blueprint, and a proof of convergence to subgame-perfect equilibrium. We implement our approach with deep RL, validate its performance empirically, and introduce a theoretical analysis in the presence of approximation errors, including a nudging mechanism that uses small extra payments to discourage potential deviations by the agent. We demonstrate the framework's effectiveness for solving sequential social dilemmas, showing how it can maximize social welfare with minimal intervention to agents' rewards. Our findings suggest several promising directions for future research, including scaling to even more complex environments, exploring partially-observable settings where different agents have different observations, and incorporating randomized contracts for enhanced agent coordination. We believe our work will inspire further exploration at the intersection of principal-agent theory and RL, potentially yielding novel solutions to complex AI agent interactions.

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

# Appendix

## Table of Contents

## A   RELATED WORK

### A.1   AUTOMATED MECHANISM DESIGN

Our study concerns finding an optimal way to influence the behavior of one or multiple agents in an environment through optimization, it can thus be attributed to the automated mechanism design literature. The field was pioneered by Conitzer & Sandholm (2002; 2003; 2004); the early approaches mostly concerned optimal auctions and relied on classic optimization and machine learning algorithms (Likhodedov et al., 2005; Lahaie, 2011; Sandholm & Likhodedov, 2015; Dütting et al., 2015; Narasimhan et al., 2016). The field has received a surge of interest with the introduction of RegretNet (Dütting et al., 2019) – a deep learning based approach to approximately incentive-compatible optimal auction design. This inspired multiple other algorithms for auction design (Rahme et al., 2021; Duan et al., 2022; Ivanov et al., 2022; Duan et al., 2023; Wang et al., 2024b), as well as applications of deep learning to other economic areas such as multi-facility location (Golowich et al., 2018), two-sided matching markets (Ravindranath et al., 2021), E-commerce advertising (Liu et al., 2021), and data markets (Ravindranath et al., 2023).

Notably, a deep learning approach to contract design has been recently proposed by Wang et al. (2023) as a viable alternative to linear programming in problems with a high number of actions and outcomes. Since our investigation focuses on scenarios where the primary source of complexity comes from large state spaces rather than action or outcome spaces, we do not use their approximation technique. At the heart of their approach is a novel neural network architecture specifically designed to approximate discontinuous functions. Given that the principal's Q-function, $Q^\rho(s, b \mid \pi)$, in our setting is discontinuous with respect to $b$, this architecture holds potential to bring further scalability and practicality to our approach; we leave this direction as future work.

## A.2 ALGORITHMIC CONTRACT DESIGN

A body of work studies repeated principal-agent interactions in games either stateless or with states unobserved by the principal (such as agent types). Depending on the model, learning an optimal payment scheme can be formalized as a bandit problem with arms representing discretized contracts (Conitzer & Garera, 2006; Ho et al., 2014; Zhu et al., 2023b) or as a constrained dynamic programming problem (Renner & Schmedders, 2020). Scheid et al. (2024) extend the bandit formulation to a linear contextual setting and propose a learning algorithm that is near-optimal in terms of principal's regret. Zhang et al. (2023) formulate a so-called steering problem where a mediator can pay no-regret learners throughout repeated interactions and wishes to incentivize some desirable predetermined equilibrium while satisfying budget constraints. Similarly, Guruganesh et al. (2024) study a repeated principal-agent interaction in a canonical contract setting, with the agent applying no-regret learning over the course of interactions. Li et al. (2021) study contracts for afforestation with an underlying Markov chain; they do not extend to MDPs and do not apply learning.

There is also work on *policy teaching*, which can be seen as the earliest examples of contract design in MDPs. Zhang et al. (2009b) study a problem of implementing a specific policy through contracts and solve it with linear programming. Zhang & Parkes (2008) additionally aim to find the policy itself, which they show to be NP-hard and solve through mixed integer programming. Contemperaneously with the present work, Ben-Porat et al. (2024) extend these results by offering polynomial approximation algorithms for two special instances of MDPs. These, as well as our work, can be seen as instances of a more general *environment design* problem (Zhang et al., 2009a). Crucially, these works focus on MDPs of up to a hundred states. By employing deep RL, we extend to much larger MDPs. Our approach also generalizes to hidden-action and multi-agent MDPs.

Monderer & Tennenholtz (2003) propose *k-implementation*, which can be seen as contract design applied to normal-form games. Specifically, the principal wants to implement (incentivize) some desirable outcome and can pay for the joint actions of the agents. The goal is to find the k-implementation (we call it a minimal implementation), i.e., such payment scheme that the desirable outcome is dominant-strategy incentive compatible for all agents, while the realized payment $k$ for this outcome is minimal. Our multi-agent problem setup can be seen as learning a minimal implementation of a social welfare maximizing strategy profile in a Markov game.

Related to algorithmic contract design is a problem of delegating learning tasks in the context of incentive-aware machine learning (Ananthakrishnan et al., 2024; Saig et al., 2023). These studies concern a principal properly incentivizing agent(s) through contracts to collect data or train an ML model in a one-shot interaction.

## A.3 MULTI-AGENT RL (MARL)

In our applications, we focus on general-sum Markov games where naively trained agents fail to engage in mutually beneficial cooperation – colloquially known as "Sequential Social Dilemmas" or SSDs (Leibo et al., 2017). The solution concepts can be divided into two broad categories. The majority of studies take a purely computational perspective, arbitrarily modifying the agents' reward functions (Peysakhovich & Lerer, 2018a;b; Hughes et al., 2018; Jaques et al., 2019; Wang et al., 2019; Eccles et al., 2019; Jiang & Lu, 2019; Durugkar et al., 2020; Yang et al., 2020; Ivanov et al., 2021; Zimmer et al., 2021; Phan et al., 2022) or training procedures (Gupta et al., 2017; Foerster et al., 2018; Willi et al., 2022; Zhao et al., 2022) in order to maximize the aggregate reward. Alternative solutions view the problem as aligning the players' incentives by modifying the rules of the game to induce better equilibria. Examples include enabling agents to delegate their decision making to a mediator (Ivanov et al., 2023), allowing agents to review each others' policies prior to decision making (Oesterheld et al., 2023), and adding a cheap-talk communication channel between agents (Lin et al., 2024). Our work should be attributed to the second category as we model the principal-agents interaction as a game. While the principal effectively modifies agents' reward functions, the payments are costly. The question is then how to maximize social welfare through minimal intervention, which is an open research question.

The works on adaptive mechanism design (Baumann et al., 2020; Guresti et al., 2023) can be seen as precursors of contract design for SSDs. These consider augmenting the game with a principal-like planning agent that learns to distribute additional rewards and penalties, the magnitude of which is

Table 2: Comparison of standard and Principal-Agent MDPs

| | MDP | Principal-Agent MDP observed action | hidden action |
|---|---|---|---|
| States | $S$ | $S$ | $S$ |
| Agent's actions ($n$ elements) | $A$ | $A$ | $A$ |
| Outcomes ($m$ elements) | – | – | $O$ |
| Principal's actions | — | $B \subset \mathbb{R}_{\geq 0}^n$ | $B \subset \mathbb{R}_{\geq 0}^m$ |
| MDP transitioning | $s' \sim \mathcal{T}(s,a)$ | $s' \sim \mathcal{T}(s,a)$ | $o \sim \mathcal{O}(s,a), s' \sim \mathcal{T}(s,o)$ |
| Agent's reward | $r(s,a)$ | $r(s,a) + b(a)$ | $r(s,a) + b(o)$ |
| Principal's reward | — | $r^p(s,a) - b(a)$ | $r^p(s,o) - b(o)$ |
| Agent's policy | $\pi(s)$ | $\pi(s,b)$ | $\pi(s,b)$ |
| Principal's policy | — | $\rho(s)$ | $\rho(s)$ |
| Agent's value | $V^\pi(s)$ | $V^\pi(s,b \mid \rho)$ | $V^\pi(s,b \mid \rho)$ |
| Principal's value | — | $V^\rho(s \mid \pi)$ | $V^\rho(s \mid \pi)$ |

either limited heuristically or handcoded. Importantly, the planning agent is not considered a player, and thus the equilibria are not analyzed.

Most relevant to us, Christoffersen et al. (2023) consider a contracting augmentation of SSDs. Before an episode begins, one of the agents proposes a zero-sum reward redistribution scheme that triggers according to predetermined conditions. Then, the other agents vote on accepting it, depending on which the episode proceeds with the original or modified rewards. Because the conditions are handcoded based on domain knowledge, the contract design problem reduces to finding a one-dimensional parameter from a discretized interval that optimizes the proposal agent's welfare, and by the symmetry of contracts, the social welfare. Besides the technicality that the principal in our setting is external to the environment, a crucial distinction is that we employ contracts in full generality, allowing the conditions for payments to emerge from learning. We empirically verify that this may result in a performance gap. To adapt this approach to the Coin Game and relax the domain knowledge assumption, we 1) duplicate parts of rewards rather than redistribute them, which can also be interpreted as payments by an external principal, and 2) allow each agent to immediately share the duplicated part of its reward with the other agent, rather than a constant value every time a handcoded condition is met. In experiments, we refer to this as a 'constant proportion baseline'.

This work only covers fully observable environments, but our method could potentially be extended to partial observability, limiting the information available to the principal and the agents to local observations. In this regard, our method may be considered as having decentralized execution. While the presence of a principal as a third party may be considered a centralized element, even this could be alleviated through the use of cryptography (Sun et al., 2023).

# B  PROOFS AND DERIVATIONS (SECTIONS 3 AND 4)

In this appendix, we provide formal proofs and derivations for the results in Sections 3 and 4. Appendix B.1 provides additional intuition on the differences between the solution concepts of Stackelberg and Subgame-Perfect Equilibria. Appendix B.2 supplements Observation 3.1 and defines the principal's and agent's MDPs. Appendix B.3 defines contraction operators useful for succeeding proofs and connects these operators to the modified Q-functions defined in Section 4. Appendices B.4 and B.5 present the respective proofs of Theorems 3.3 and 3.4. Whereas the theory in the main text focuses on the finite-horizon hidden-action scenario, we also discuss the infinite-horizon and observed-action scenarios in appendices B.6 and B.7, respectively. The linear program formulated in Section 4 is derived and described in more detail in Appendix B.8. Finally, in Appendix B.9 we analyze convergence of the meta-algorithm in the presence of approximation errors, focusing on error propagation and utility guarantees, and propose the nudging method.

B.0   COMPARISON OF STANDARD AND PRINCIPAL-AGENT MDPS

Table 2 summarizes the differences between standard MDPs and Principal-Agent MDPs with and without hidden actions.

B.1   STACKELBERG VS SUBGAME-PERFECT EQUILIBRIUM (EXAMPLE IN FIGURE 1, REVISITED)

First, consider the SPE notion: At the left subgame, the principal incentivizes the agent to take noisy-left by choosing a contract that pays 1 for outcome $L$ and 0 otherwise. This way, both actions yield the same value for the agent, $0.9 \cdot 1 + 0.1 \cdot 0 - 0.8 = 0.1 \cdot 1 + 0.9 \cdot 0 = 0.1$, and the agent chooses $a_L$ by tie-breaking in favour of the principal. So the principal's value in state $s_L$ is $0.9(r^p(s_L, L) - 1) = 0.9(\frac{14}{9} - 1) = 0.5$. By the same logic, the principal offers the same contract in $s_R$ and its value equals $0.5$ (and the agent's value equals $0.1$). Then, in $s_0$, the agent is indifferent between transitioning to $s_L$ and $s_R$, so the principal has to offer the same contract again. The principal's value in $s_0$ (given that the agent chooses $a_L$) is $0.5 + 0.9 \cdot 0.5 + 0.1 \cdot 0.5 = 1$, and the agent's value is $0.1 + 0.9 \cdot 0.1 + 0.1 \cdot 0.1 = 0.2$. These are estimated by considering utilities in $s_L$ and $s_R$ without discounting (using $\gamma = 1$). Note that in this analysis, we found SPE using backward induction: we first analyzed the terminal states, then the root state.

Compare this with Stackelberg: If non-credible threats were allowed, the principal could threaten to act suboptimally in the right subgame by always paying the agent 0. Both principal and agent then value $s_R$ at 0. In $s_L$, the contract is the same as in SPE. Knowing this, the agent values $s_L$ over $s_R$ ($0.1 > 0$), which would drive it to choose the noisy-left action at the root state even if the principal pays only $1 - 0.1 = 0.9$ for outcome $L$. By paying less, the principal's utility (value in $s_0$) would be higher compared to SPE, $(\frac{14}{9} - 0.9 + 0.5) \cdot 0.9 = 1.04 > 1$.

This illustrates how Stackelberg equilibrium may produce more utility for the principal, but the surplus comes at the cost of the inability to engage in mutually beneficial contractual agreements in certain subgames, even if these subgames are reached by pure chance and despite the agent's best efforts. On the one hand, Stackelberg requires more commitment power from the principal. Whereas in SPE the agent can be certain that the principal sticks to its policy in future states because it is optimal in any subgame, in Stackelberg, the principal has to preemptively commit to inefficient contracts and ignore potential beneficial deviations. On the other hand, Stackelberg is not robust to mistakes: if the agent mistakenly chooses the wrong action, it might be punished by the principal, losing utility for both players. This is especially concerning in the context of learning, where mistakes can happen due to approximation errors, or even due to the agent exploring (in online setups). SPE is hence a more practical solution concept.

B.2   PRINCIPAL'S AND AGENT'S MDPS (OBSERVATION 3.1)

A (standard) MDP is a tuple $(S, S_0, A, \mathcal{R}, \mathcal{T}, \gamma)$, where $S$ is a set of states, $S_0 \in S$ is a set of possible initial states, $A$ is a set of actions, $\mathcal{T} : S \times A \to \Delta(S)$ is a stochastic transition function, $\mathcal{R} : S \times A \times S \to \mathcal{P}(\mathbb{R})$ is a stochastic reward function ($\mathcal{R}(s, a, s')$ is a distribution and may depend on the next state $s'$), $\gamma$ is an (optional) discounting factor. Assume finite horizon (as in Section 2).

Consider a hidden-action principal-agent MDP $\mathcal{M} = (S, s_0, A, B, O, \mathcal{O}, \mathcal{R}, \mathcal{R}^p, \mathcal{T}, \gamma)$ as defined in Section 2. First, consider the agent's perspective. Let the principal's policy be some $\rho$. This defines a standard MDP $(S^a, S_0^a, A, \mathcal{R}^a, \mathcal{T}^a, \gamma)$ that we call the agent's MDP, where: The set of states is $S^a = S \times B$ (infinite unless $B$ is discretized). The set of initial states is $S_0 = \{s_0\} \times B$. The set of actions is $A$.

The transition function $\mathcal{T}^a : S^a \times A \to \Delta(S^a)$ defines distributions over next state-contract pairs $(s', \rho(s'))$, where $s' \sim \mathcal{T}(s, o \sim \mathcal{O}(s, a))$. Note that the next state $s'$ is sampled from a distribution that is a function of state-action pairs $(s, a)$ marginalized over outcomes, and the next contract $\rho(s')$ is given by the principal's policy. Informally, the agent expects the principal to stick to its policy in any future state. At the same time, since the state space $S^a$ is defined over the set of contracts $B$, the agent may adapt its policy to any immediate contract in the current state, and thus its policy $\pi : S^a \to \Delta(A)$ is defined by $\pi(s, b)$ for any $b \in B$.

The reward function is a bit cumbersome to formalize because it should not depend on outcomes, while both $\mathcal{O}$ and $\mathcal{T}$ can be stochastic. Specifically, the new reward function has to be stochastic w.r.t. outcomes: $\mathcal{R}^a((s,b),a,s') = [r(s,a) + b(o) \mid o \sim P(o \mid s,a,s')]$, where the conditional distribution of outcomes is given by $P(o \mid s,a,s') = \frac{P(\mathcal{O}(s,a)=o)P(\mathcal{T}(s,o)=s')}{\sum_{o^*} P(\mathcal{O}(s,a)=o^*)P(\mathcal{T}(s,o^*)=s')}$.

Next, consider the principal's perspective. Let the agent's policy be some $\pi$. This defines a standard MDP $(S, \{s_0\}, B, \mathcal{R}^p, \mathcal{T}^p, \gamma)$ that we call the principal's MDP, where: the set of states is $S$; the set of initial states is $\{s_0\}$; the set of actions is $B$ (infinite unless discretized); the transition function is defined by $\mathcal{T}^p(s,b) = \mathcal{T}(s, o \sim \mathcal{O}(s, a \sim \pi(s,b)))$; the reward function is defined similarly to the agent's MDP by $\mathcal{R}^p(s,b,s') = [r(s,o) - b(o) \mid o \sim P(o \mid s,a,s')]$.

Thus, the optimization task of the principal (agent) given a fixed policy of the agent (principal) can be cast as a standard single-agent MDP. In both players' MDPs, the other player's presence is implicit, embedded into transition and reward functions. Note that our definition of standard MDP allows for stochastic reward functions that depend on the next state, as well as a set of initial states that is not a singleton. The same generalizations can be made in our Principal-Agent MDP definition, and in particular, the above derivations would still hold, but we decided to slightly specify our model for brevity.

## B.3   CONTRACTION OPERATORS AND THEIR FIXED POINTS

Our meta-algorithm iteratively solves a sequence of principal's and agent's MDPs as defined in Appendix B.2. Both tasks can be performed with Q-learning and interpreted as finding fixed points of the Bellman optimality operator in the respective MDPs. Furthermore, our learning approach in Section 4 makes use of modified Q-functions, which are also fixed points of contraction operators. For convenience, we define all four operators below. Where necessary, we prove the operators being contractions and define their fixed points.

### B.3.1   OPTIMALITY OPERATORS IN THE AGENT'S MDP

Here we assume some fixed principal's policy $\rho$ that defines an agent's MDP.

**Bellman optimality operator.**   Consider the agent's optimization task (line 3 of the meta-algorithm). Solving it implies finding a fixed point of an operator $\mathcal{S}_\rho$, which is the Bellman optimality operator in the agent's MDP defined by a principal's policy $\rho$.

**Definition B.1.** Given an agent's MDP defined by a principal's policy $\rho$, the Bellman optimality operator $\mathcal{S}_\rho$ is defined by

$$(\mathcal{S}_\rho Q)((s,b),a) = \mathbb{E}_{o \sim \mathcal{O}(s,a), s' \sim \mathcal{T}(s,o)}\Big[\big(\mathcal{R}(s,a,b,o) + \gamma \max_{a'} Q((s', \rho(s')), a')\big)\Big], \qquad (6)$$

where $\mathcal{R}(s,a,b,o) = r(s,a) + b(o)$, $Q$ is an element of a vector space $\mathcal{Q}_{SBA} = \{S \times B \times A \to \mathbb{R}\}$, and subscript $\rho$ denotes conditioning on the principal's policy $\rho$.

This operator is a contraction and admits a unique fixed point $Q^*(\rho)$ that satisfies:

$$V^*(s,b \mid \rho) = \max_a Q^*((s,b), a \mid \rho), \qquad (7)$$

$$Q^*((s,b), a \mid \rho) = \mathbb{E}\Big[\mathcal{R}(s,a,b,o) + \gamma \max_{a'} Q^*((s', \rho(s')), a' \mid \rho)\Big]. \qquad (8)$$

The policy corresponding to the fixed point is called the best-responding policy $\pi^*(\rho)$:

$$\forall s \in S, b \in B : \pi^*(s,b \mid \rho) = \arg\max_a Q^*((s,b), a \mid \rho),$$

where ties are broken in favour of the principal.

**Truncated Bellman optimality operator.** Here we show that the truncated Q-function $\overline{Q}^*(\rho)$ defined in the main text (1) is a fixed point of a contraction operator and thus can be found with Q-learning.

**Definition B.2.** Given an agent's MDP defined by principal's policy $\rho$, the truncated Bellman optimality operator $\overline{\mathcal{S}}_\rho$ is defined by

$$(\overline{\mathcal{S}}_\rho \overline{Q})(s,a) = r(s,a) + \gamma \mathbb{E}_{o\sim\mathcal{O}(s,a), s'\sim\mathcal{T}(s,o)} \max_{a'}\big[\mathbb{E}_{o'\sim\mathcal{O}(s',a')}\rho(s')(o') + \overline{Q}(s',a')\big], \quad (9)$$

where $\overline{Q}$ is an element of a vector space $\mathcal{Q}_{SA} = \{S \times A \to \mathbb{R}\}$.

**Lemma B.3.** *Operator $\overline{\mathcal{S}}_\rho$ is a contraction in the sup-norm.*[3]

*Proof.* Let $\overline{Q}_1, \overline{Q}_2 \in \mathcal{Q}_{SA}, \gamma \in [0,1)$. The operator $\overline{\mathcal{S}}_\rho$ is a contraction in the sup-norm if it satisfies $\|\overline{\mathcal{S}}_\rho \overline{Q}_1 - \overline{\mathcal{S}}_\rho \overline{Q}_2\|_\infty \leq \gamma\|\overline{Q}_1 - \overline{Q}_2\|_\infty$. This inequality holds because:

$$\|\overline{\mathcal{S}}_\rho \overline{Q}_1 - \overline{\mathcal{S}}_\rho \overline{Q}_2\|_\infty = \max_{s,a}\Big|\gamma\mathbb{E}_{o,s'}\big[\max_{a'}(\mathbb{E}_{o'}\rho(s')(o') + \overline{Q}_1(s',a'))-$$
$$\max_{a'}(\mathbb{E}_{o'}\rho(s')(o') + \overline{Q}_2(s',a'))\big] + r(s,a) - r(s,a)\Big| \leq$$
$$\max_{s,a}\Big|\gamma\mathbb{E}_{o,s'} \max_{a'} \mathbb{E}_{o'}\big[\rho(s')(o') + \overline{Q}_1(s',a') - \rho(s')(o') - \overline{Q}_2(s',a')\big]\Big| =$$
$$\max_{s,a}\Big|\gamma\mathbb{E}_{o,s'} \max_{a'}\big[\overline{Q}_1(s',a') - \overline{Q}_2(s',a')\big]\Big| \leq$$
$$\max_{s,a} \gamma \max_{s',a'}\big|\overline{Q}_1(s',a') - \overline{Q}_2(s',a')\big| = \gamma\|\overline{Q}_1 - \overline{Q}_2\|_\infty.$$

$\square$

Because $\overline{\mathcal{S}}_\rho$ is a contraction as shown in Lemma B.3, by the Banach theorem, it admits a unique fixed point $\overline{Q}^*_\rho$ s.t. $\forall s,a : \overline{Q}^*_\rho(s,a) = (\overline{\mathcal{S}}_\rho \overline{Q}^*_\rho)(s,a)$. We now show that this fixed point is the truncated Q-function. Define $Q_\rho((s,b),a) = \mathbb{E}_{o\sim\mathcal{O}(s,a)}b(o) + \overline{Q}^*_\rho(s,a)$. Notice that the fixed point satisfies:

$$\forall s \in S, a \in A : \overline{Q}^*_\rho(s,a) = (\overline{\mathcal{S}}_\rho \overline{Q}^*_\rho)(s,a) \overset{(Eq.\ 9)}{=}$$
$$r(s,a) + \gamma\mathbb{E}_{o,s'} \max_{a'}\big[\mathbb{E}_{o'}\rho(s')(o') + \overline{Q}^*_\rho(s',a')\big] =$$
$$r(s,a) + \gamma\mathbb{E}_{o,s'} \max_{a'} Q_\rho((s',\rho(s')),a').$$

At the same time, by definition:

$$\forall s \in S, a \in A : \overline{Q}^*_\rho(s,a) = Q_\rho((s,b),a) - \mathbb{E}_{o\sim\mathcal{O}(s,a)}b(o).$$

Combining the above two equations and swapping terms:

$$\forall s \in S, a \in A : Q_\rho((s,b),a) = r(s,a) + \mathbb{E}_{o,s'}[b(o) + \gamma \max_{a'} Q_\rho((s',\rho(s')),a')].$$

Notice that the last equation shows that $Q_\rho$ is the fixed point of the Bellman optimality operator $S_\rho$ (6), i.e., $Q_\rho \equiv Q^*(\rho)$, as it satisfies the optimality equations (8). It follows that $Q^*((s,b),a \mid \rho) = \mathbb{E}_o b(o) + \overline{Q}^*_\rho(s,a)$, and thus $\overline{Q}^*_\rho$ satisfies the definition of the truncated Q-function (1), i.e., $\overline{Q}^*_\rho \equiv \overline{Q}^*(\rho)$. The truncated Q-function is then a fixed point of a contraction operator and can be found with Q-learning. It can also be used to compute the best-responding policy: $\pi^*(s,b \mid \rho) = \arg\max_a[\mathbb{E}_o b(o) + \overline{Q}^*(s,a \mid \rho)]$.

---

[3]In lemmas B.3 and B.6, we show for a case that includes finite- and infinite-horizon MDPs, but requires $\gamma < 1$. For $\gamma = 1$ and finite-horizon MDPs, per-timestep operators can be shown to be contractions, similar to the Bellman operator in chapter 4.3 of Puterman (2014).

B.3.2 OPTIMALITY OPERATORS IN THE PRINCIPAL'S MDP

Here we assume some fixed best-responding agent's policy $\pi^*(\rho)$ that defines a principal's MDP. While we could instead assume an arbitrary policy $\pi$, we are only interested in solving the principal's MDP as the outer level of the meta-algorithm, which always follows the inner level that outputs an agent's policy $\pi^*(\rho)$ best-responding to some $\rho$.

**Bellman optimality operator.**   Consider the principal's optimization level (line 4 of the meta-algorithm). Solving it implies finding a fixed point of an operator $\mathcal{B}_\rho$, which is the Bellman optimality operator in the principal's MDP defined by the agent's policy $\pi^*(\rho)$ is best-responding to some $\rho$.

**Definition B.4.** Given a principal's MDP defined by the agent's policy $\pi^*(\rho)$ best-responding to some $\rho$, the Bellman optimality operator $\mathcal{B}_\rho$ is defined by

$$(\mathcal{B}_\rho Q)(s,b) = \mathbb{E}_{o \sim \mathcal{O}(s,a), s' \sim \mathcal{T}(s,o)}\Big[\big(\mathcal{R}^p(s,b,o) + \gamma \max_{b'} Q(s',b')\big) \mid a = \pi^*(s,b \mid \rho)\Big], \quad (10)$$

where $\mathcal{R}^p(s,b,o) = r^p(s,o) - b(o)$, $Q$ is an element of a vector space $\mathcal{Q}_{SB} = \{S \times B \to \mathbb{R}\}$, and subscript $\rho$ denotes conditioning on the agent's best-responding policy $\pi^*(\rho)$.

This operator is a contraction and admits a unique fixed point $Q^*(\pi^*(\rho))$ that satisfies optimality equations:

$$V^*(s \mid \pi^*(\rho)) = \max_b Q^*(s,b \mid \pi^*(\rho)), \quad (11)$$

$$Q^*(s,b \mid \pi^*(\rho)) = \mathbb{E}\Big[\big(\mathcal{R}^p(s,b,o) + \gamma \max_{b'} Q^*(s',b' \mid \pi^*(\rho))\big) \mid a = \pi^*(s,b \mid \rho)\Big]. \quad (12)$$

The policy corresponding to the fixed point is called the subgame-perfect policy $\rho^*(\pi^*(\rho))$:

$$\forall s \in S : \rho^*(s \mid \pi^*(\rho)) = \arg\max_b Q^*(s,b \mid \pi^*(\rho)).$$

**Contractual Bellman optimality operator.**   Here we show that the contractual Q-function $q^*(\pi^*(\rho))$ defined in the main text (2) is a fixed point of a contraction operator and thus can be found with Q-learning.

**Definition B.5.** Given a principal's MDP defined by the agent's policy $\pi^*(\rho)$ best-responding to some $\rho$, the contractual Bellman optimality operator $\mathcal{H}_\rho$ is defined by

$$(\mathcal{H}_\rho q)(s,a^p) = \max_{\{b \mid \pi^*(s,b \mid \rho) = a^p\}} \mathbb{E}_{o \sim \mathcal{O}(s,a^p), s' \sim \mathcal{T}(s,o)}\Big[\mathcal{R}^p(s,b,o) + \gamma \max_{a'} q(s',a')\Big], \quad (13)$$

where $q$ is an element of a vector space $\mathcal{Q}_{SA} = \{S \times A \to \mathbb{R}\}$, and $a^p \in A$ denotes the principal's *recommended* action.

**Lemma B.6.** *Operator $\mathcal{H}_\rho$ is a contraction in the sup-norm.*

*Proof.* Let $q_1, q_2 \in \mathcal{Q}_{SA}$, $\gamma \in [0,1)$. The operator $\mathcal{H}_\rho$ is a contraction in the sup-norm if it satisfies $\|\mathcal{H}_\rho q_1 - \mathcal{H}_\rho q_2\|_\infty \leq \gamma \|q_1 - q_2\|_\infty$. This inequality holds because:

$$\|\mathcal{H}_\rho q_1 - \mathcal{H}_\rho q_2\|_\infty = \max_{s,a}\Big| \max_{\{b \in B \mid \pi^*(s,b \mid \rho) = a\}} \mathbb{E}\big[\mathcal{R}^p(s,b,o) + \gamma \max_{a'} q_1(s',a')\big] -$$

$$\max_{\{b \in B \mid \pi^*(s,b \mid \rho) = a\}} \mathbb{E}\big[\mathcal{R}^p(s,b,o) + \gamma \max_{a'} q_2(s',a')\big]\Big| =$$

$$\max_{s,a} \gamma \Big| \mathbb{E}[\max_{a'} q_1(s',a') - \max_{a'} q_2(s',a')]\Big| \leq$$

$$\max_{s,a} \gamma \mathbb{E}\Big| \max_{a'} q_1(s',a') - \max_{a'} q_2(s',a')\Big| \leq$$

$$\max_{s,a} \gamma \mathbb{E} \max_{a'} |q_1(s',a') - q_2(s',a')| \leq$$

$$\max_{s,a} \gamma \max_{s',a'} |q_1(s',a') - q_2(s',a')| = \gamma \|q_1 - q_2\|_\infty.$$

$\square$

Because $\mathcal{H}_\rho$ is a contraction as shown in Lemma B.6, by the Banach theorem, it admits a unique fixed point $q_\rho^*$ s.t. $\forall s, a^p : q_\rho^*(s, a^p) = (\mathcal{H}_\rho q_\rho^*)(s, a^p)$. We now show that this fixed point is the contractual Q-function. Notice that the fixed point satisfies:

$$
\begin{aligned}
\forall s \in S : \ \max_{a^p} q_\rho^*(s, a^p) &= \max_{a^p}(\mathcal{H}_\rho q_\rho^*)(s, a^p) \overset{(Eq.\ 13)}{=} \\
\max_b (\mathcal{H}_\rho q_\rho^*)(s, \pi^*(s, b \mid \rho)) &= \max_b(\mathcal{H}_\rho(\mathcal{H}_\rho q_\rho^*))(s, \pi^*(s, b \mid \rho)) = \cdots = \\
\max_b \mathbb{E} \sum_t \Big[ \gamma^t \mathcal{R}^p(s_t, b_t, o_t) &\mid s_0 = s, b_0 = b, \pi^*(\rho) \Big] = \max_b Q^*(s, b \mid \pi^*(\rho)),
\end{aligned}
\tag{14}
$$

and:

$$
\begin{aligned}
\forall s \in S, a^p \in A : \ q_\rho^*(s, a^p) &= (\mathcal{H}_\rho q_\rho^*)(s, a^p) \overset{(Eq.\ 13)}{=} \\
\max_{\{b \mid \pi^*(s,b\mid\rho)=a^p\}} \mathbb{E}\Big[ \mathcal{R}^p(s, b, o) &+ \gamma \max_{a'} q_\rho^*(s', a') \Big] \overset{(Eq.\ 14)}{=} \\
\max_{\{b \mid \pi^*(s,b\mid\rho)=a^p\}} \mathbb{E}\Big[ \mathcal{R}^p(s, b, o) &+ \gamma \max_{b'} Q^*(s', b' \mid \pi^*(\rho)) \Big] \overset{(Eq.\ 12)}{=} \\
\max_{\{b \mid \pi^*(s,b\mid\rho)=a^p\}} Q^*(s, b \mid \pi^*(\rho)). &
\end{aligned}
\tag{15}
$$

Thus, $q_\rho^*$ satisfies the definition of the contractual Q-function (2), i.e., $q_\rho^* \equiv q^*(\pi^*(\rho))$. The contractual Q-function is then a fixed point of a contraction operator and can be found with Q-learning. The contractual Q-function can also be used to compute the subgame-perfect principal's policy as $\rho^*(s) = \arg\max_b(\mathcal{H}_\rho q_\rho^*)(s, \pi^*(s, b \mid \rho)) = \arg\max_b Q^*(s, b \mid \pi^*(\rho))$. We address computing the $\arg\max$ in Appendix B.8 using Linear Programming.

### B.4 META-ALGORITHM FINDS SPE (PROOF OF THEOREM 3.3)

**Observation B.7.** A principal-agent stochastic game $\mathcal{G}$ is in SPE if and only if every subgame of $\mathcal{G}$ is in SPE.

Observation B.7 will be useful for the proofs in this section.

Let $S_t \subseteq S$ denote the set of all possible states at time step $t$. For example, $S_0 = \{s_0\}$. States in $S_T$ are terminal by definition. We assume that sets $S_t$ are disjoint. This is without loss of generality, as we can always redefine the state space of a finite-horizon MDP such that the assumption holds; e.g., by concatenating the time step to a state as $s_t^{new} = (s_t, t)$. In other words, any finite-horizon MDP can be represented as a directed acyclic graph.

The next lemma is a precursor to proving the convergence of Algorithm 1 and concerns its single iteration. Given a pair of policies that form an SPE in all subgames but the original game, it states that performing one additional iteration of the algorithm (update the agent, then the principal) yields an SPE in the game. This is because the agent observes the contract offered in a state and adapts its action, in accordance with our definition of the agent's MDP in Appendix B.2. In particular, the agent's best-responding policy is defined as $\pi^*(s, b \mid \rho)$ for *any* pair $(s, b)$, so in $s_0$, the agent best-responds with $\pi^*(s_0, b \mid \rho)$ for any $b$ regardless of the principal's policy $\rho(s_0)$.

**Lemma B.8.** *Given a finite-horizon principal-agent stochastic game $\mathcal{G}$ and a principal's policy $\rho$, if $(\rho, \pi^*(\rho))$ is an SPE in all subgames with a possible exception of $\mathcal{G}$ (i.e., all subgames in $s \in S \setminus \{s_0\}$), then $(\rho^*(\pi^*(\rho)), \pi^*(\rho))$ is an SPE in $\mathcal{G}$.*

*Proof.* In $s_0$, the agent observes the principal's action. Consequently and because the agent best-responds to $\rho$, it also best-responds to any $\{\rho' \mid \forall s \in S \setminus \{s_0\} : \rho'(s) = \rho(s)\}$ regardless of the offered contract $\rho'(s_0)$, including the subgame-perfect $\rho^*(\pi^*(\rho))$ (that only differs from $\rho$ in $s_0$, as subgames in other states are in SPE already). Thus, $(\rho^*(\pi^*(\rho)), \pi^*(\rho))$ is an SPE in $\mathcal{G}$, as well as in all subgames of $\mathcal{G}$ (by Observation B.7). $\square$

**Corollary B.9.** *Given $\mathcal{G}$ with a single subgame ($S = \{s_0\}$), $(\rho^*(\pi^*(\rho)), \pi^*(\rho))$ is an SPE in $\mathcal{G}$ for any $\rho$.*

This corollary concerns MDPs with a single state, which could also be interpreted as stateless. This covers static contract design problems from Section 2.1, as well as subgames in terminal states of Principal-Agent MDPs.

By using Lemma B.8, we can now show that each iteration of the algorithm expands the set of subgames that are in SPE, as long as some subgames satisfy the conditions of the lemma for an arbitrarily initialized $\rho$. This is always the case for finite-horizon MDPs, as the subgames in terminal states have a single subgame (itself), and thus Corollary B.9 applies. This reasoning is used to prove Theorem 3.3.

*Proof of Theorem 3.3.* Denote the policy initialized at line 1 as $\rho_0$. Denote the best-responding policy to $\rho_0$ as $\pi_1 \equiv \pi^*(\rho_0)$ and the subgame-perfect policy against $\pi_1$ as $\rho_1 \equiv \rho^*(\pi_1)$. Likewise, $\pi_i$ and $\rho_i$ respectively denote the best-responding policy to $\rho_{i-1}$ and the subgame-perfect policy against $\pi_i$, where $i$ is an iteration of the algorithm.

By Corollary B.9, $(\rho_1, \pi_1)$ forms an SPE in all subgames in terminal states, including $s \in S_T$. Applying Lemma B.8, as well as our w.l.o.g. assumption that sets $S_t$ are disjoint, $(\rho_2, \pi_2)$ is an SPE in all subgames in $S_T \cup S_{T-1}$. By induction, $(\rho_i, \pi_i)$ is an SPE in all subgames in $s \in S_T \cup S_{T-1} \cdots \cup S_{T-i+1}$. Thus, $(\rho_{T+1}, \pi_{T+1})$ is an SPE in all subgames in $s \in S_T \cup \cdots \cup S_0 = S$, and thus in the game $\mathcal{G}$ (by Observation B.7). $\qquad\square$

*Remark B.10.* The above proof shows that the meta-algorithm implicitly performs backward induction. In particular, applying backward induction to an extensive-form game (which a finite-horizon Principal-Agent game instantiates) involves solving the game tree one level at a time, starting with leaves and ending with root. Similarly, the meta-algorithm solves one level at a time (subgames in states from $S_t$ at an iteration $t$), but does so implicitly, as a consequence of updating the players' policies (in all states). This distinction has far-reaching implications, and in particular, allows us to employ model-free RL.

B.5   META-ALGORITHM APPLIES CONTRACTION (PROOF OF THEOREM 3.4)

Consider the principal's optimization task (line 4 of the meta-algorithm). As we discuss in Appendix B.3, solving this task can be interpreted as finding the fixed point of a contraction operator $\mathcal{B}_\rho$ defined in (10). By definition of contraction, this fixed point can be found by iteratively applying the operator until convergence, which we denote as $\mathcal{H}^* = \mathcal{B}_\rho(\mathcal{B}_\rho(\mathcal{B}_\rho...Q(s,b)))$.

**Observation B.11.** $\mathcal{H}^*$ is a composition of linear operators and thus is a linear operator.

For a linear operator $\mathcal{H}^*$ to be a contraction, its fixed point has to be unique. Since its iterative application (the meta-algorithm, Algorithm 1) converges to an SPE, we next prove the uniqueness of Q-functions in SPE.

**Lemma B.12.** *Given a finite-horizon principal-agent stochastic game $\mathcal{G}$, the principal's Q-function is equal in all SPE for any state-action; the same holds for the agent's Q-function.*

*Proof.* Consider a principal's policy $\rho$ that forms SPE with any best-responding agent's policy $\pi^*(\rho)$. Any $\pi^*(\rho)$ solves the agent's MDP and thus all such policies define a unique agent's Q-function (which is the fixed point of the Bellman optimality operator). Furthermore, by the assumption that the agent breaks ties in favor of the principal, all best-responding policies also uniquely define the principal's Q-function (in other words, any policy that solves the agent's MDP but does not maximize the principal's Q-function when breaking ties in some state is not a best-responding policy by the assumed tie-breaking). Thus, for any pair of SPE with non-equal Q-functions, the principal's policies must also differ.

Consider two principal's policies, $\rho_x$ and $\rho_y$, that form SPE with any respective best-responding agents' policies, $\pi^*(\rho_x)$ and $\pi^*(\rho_y)$. By the above argument, the choice of $\pi^*(\rho_x)$ and $\pi^*(\rho_y)$ is inconsequential, so we can assume those to be unique (e.g. by adding lexicographic tie-breaking if the principal-favored tie-breaking does not break all ties).

For $\rho_x$ and $\rho_y$ to differ, there must be a state $s \in S_t$ such that 1) all subgames in states "after" $t$, i.e., $\{S_{t'}\}_{t'>t}$, are in unique SPE given by some $\rho$ and $\pi^*(\rho)$ (e.g., this holds in a terminal state) and 2) the subgame in $s$ has two contracts, $b_x = \rho_x(s)$ and $b_y = \rho_y(s)$, that both maximize the principal's utility, i.e., $Q^*(s, b_x \mid \pi^*(\rho_x)) = Q^*(s, b_y \mid \pi^*(\rho_y))$. Denote the agent's actions in $s$ as $a_x = \pi^*(s, b_x \mid \rho)$ and $a_y = \pi^*(s, b_y \mid \rho)$. We now show that the choice between $b_x$ and $b_y$ is inconsequential as both contracts also yield the same utility for the agent.

Assume the agent prefers $b_x$ to $b_y$, i.e., $Q^*((s, b_x), a_x \mid \rho) > Q^*((s, b_y), a_y \mid \rho)$. First, use the observed-action notation. Applying the Bellman optimality operator and using the definition of $\mathcal{R}$, we have $\mathbb{E}[r(s, a_x) + \gamma \max_{a'} Q^*((s', \rho(s')), a' \mid \rho)] + b_x(a_x) > \mathbb{E}[r(s, a_y) + \gamma \max_{a'} Q^*((s', \rho(s')), a' \mid \rho)] + b_y(a_y)$. Observe that the principal may simply decrease the payment $b_x(a_x)$ by the difference of the agent's Q-values (so that the inequality becomes equality), increasing the principal's utility. So, the assumption that the agent prefers $b_x$ to $b_y$ means that neither contract maximizes the principal's utility in the subgame, leading to a contradiction.

The same can be shown in the hidden-action model. The agent preferring $b_x$ to $b_y$ would mean $\mathbb{E}[r(s, a_x) + b_x(o) + \gamma \max_{a'} Q^*((s', \rho(s')), a' \mid \rho)] > \mathbb{E}[r(s, a_y) + b_y(o) + \gamma \max_{a'} Q^*((s', \rho(s')), a' \mid \rho)]$, and the principal would be able to adjust $b_x$ in order to decrease the expected payment $\mathbb{E}[b_x(o)]$ relatively to $\mathbb{E}[b_y(o)]$, e.g., by decreasing each non-zero payment $b_x(o)$ by a constant. Again, this leads to a contradiction.

We thus have shown that the choice between $b_x$ and $b_y$ in $s$ is inconsequential for the value functions of both principal and agent in $s$: $V^*(s \mid \pi^*(\rho_x)) = Q^*(s, b_x \mid \pi^*(\rho_x)) = Q^*(s, b_y \mid \pi^*(\rho_y)) = V^*(s \mid \pi^*(\rho_y))$ and $V^*(s, b_x \mid \rho) = Q^*((s, b_x), a_x \mid \rho) = Q^*((s, b_y), a_y \mid \rho) = V^*(s, b_y \mid \rho)$. By Bellman optimality equations ((7) and (8) for the agent, (11) and (12) for the principal), it is also inconsequential for the players' Q-functions in all states "before" $t$, i.e., $\{S_{t'}\}_{t'<t}$. For states "after" $t$, the choice also has no effect by the MDP being finite-horizon (and our w.l.o.g. assumption about the uniqueness of states). This holds for any such $b_x$ and $b_y$ in any $s$. Thus, for any SPE $(\rho, \pi^*(\rho))$, each player's Q-function is identical in all SPE for any state-action. $\square$

*Proof of Theorem 3.4.* By Observation B.11, $\mathcal{H}^*$ is a linear operator. Moreover, as Algorithm 1 converges to SPE by Theorem 3.3 and the payoffs in SPE are unique by Lemma B.12, the iterative application of $\mathcal{H}^*$ converges to a unique fixed point. By using a converse of the Banach theorem (Theorem 1 in Daskalakis et al. (2018)), this operator is a contraction under any norm that forms a complete and proper metric space. This includes the sup-norm. $\square$

### B.6 META-ALGORITHM MAY DIVERGE IN INFINITE-HORIZON MDPS

Here we present an example of a hidden-action infinite-horizon principal-agent MDP where the meta-algorithm diverges by getting stuck in a cycle. To solve the principal's and agent's optimization tasks, we specifically developed exact solvers for principal's and agent's MDPs.

The MDP consists of two states, $s_1$ and $s_2$. In each state, the agent has two actions, $a_1$ and $a_2$, which determine probabilities of sampling one of two outcomes, $o_1$ and $o_2$. When agent chooses $a_1$ in any state $s$, outcomes are sampled with respective probabilities $\mathcal{O}(o_1 \mid s, a_1) = 0.9$ and $\mathcal{O}(o_2 \mid s, a_1) = 0.1$. Vice versa, choosing $a_2$ in any state $s$ samples an outcome with probabilities $\mathcal{O}(o_1 \mid s, a_2) = 0.1$ and $\mathcal{O}(o_2 \mid s, a_2) = 0.9$. After sampling an outcome $o_i$, the MDP deterministically transitions to $s_i$ (e.g., if $o_1$ is sampled, the MDP transitions to $s_1$ regardless of the old state and the agent's action). Choosing an action that is more likely to change the state of the MDP (so, $a_2$ in $s_1$ and $a_1$ in $s_2$) requires effort from the agent, respectively costing $c(s_1, a_2) = 1$ and $c(s_2, a_1) = 2$. The other action is free for the agent: $c(s_1, a_1) = 0$ and $c(s_2, a_2) = 0$. Other things equal, the principal prefers the agent to invest an effort: it only enjoys a reward whenever the MDP transitions to a different state, equal to $r^p(s_1, o_2) = r^p(s_2, o_1) = 1.5$. The discount factor is set to $\gamma = 0.9$.

We now describe several iterations of the meta-algorithm, showing that it oscillates between two pairs of players' policies. We report the agent's truncated Q-function (1) and the principal's contractual Q-function (2) at each iteration of the algorithm (rounded to three decimals). We also verify that these Q-functions are indeed fixed points of respective operators and thus solve the respective MDPs, but only do so in $(s_1, a_2)$ as derivations in other state-action pairs are identical.

**Initialization**

Initialize the principal with a policy $\rho_0$ that does not offer any payments, i.e., $\rho_0(s_1) = \rho_0(s_2) = (0,0)$, where we denote a contract by a tuple $b = (b(o_1), b(o_2))$.

**Iteration 1: agent**

The agent's best-responding policy $\pi_1$ simply minimizes costs by never investing an effort: $\pi_1(s_1, \rho_0(s_1)) = a_1$, $\pi_1(s_2, \rho_0(s_2)) = a_2$. This corresponds to the following truncated Q-function: $\overline{Q}^{\pi_1}(s_1, a_1) = \overline{Q}^{\pi_1}(s_2, a_2) = 0$, $\overline{Q}^{\pi_1}(s_1, a_2) = -c(s_1, a_2) = -1$ and $\overline{Q}^{\pi_1}(s_2, a_1) = -c(s_2, a_1) = -2$.

To verify that this Q-function is a fixed point of the truncated Bellman optimality operator (9), observe that the optimality equations hold in all state-action pairs. For example, in $(s_1, a_2)$ we have:

$$
\begin{aligned}
-1 = \overline{Q}^{\pi_1}(s_1, a_2) &= -c(s_1, a_2) + \gamma \mathbb{E}_{o, s', o'}[\rho(s')(o') + \max_{a'} \overline{Q}^{\pi_1}(s', a')] = \\
&-1 + 0.9[0.1 \cdot 0 + 0.9 \cdot 0] = -1.
\end{aligned}
$$

In the absence of contracts, the agent's truncated Q-function is equal to its Q-function under the principal's policy: $Q^{\pi_1}((s, \rho_0(s)), a) = \overline{Q}^{\pi_1}(s, a)$.

**Iteration 1: principal**

The principal's subgame-perfect policy $\rho_1$ attempts to incentivize effort in $s_1$ and offers the following contracts: $\rho_1(s_1) = (0, 1.25)$, $\rho_1(s_2) = (0, 0)$. This corresponds to the following contractual Q-function: $q^{\rho_1}(s_1, a_1) = 1.991$, $q^{\rho_1}(s_1, a_2) = 2.048$, $q^{\rho_1}(s_2, a_1) = 1.391$, and $q^{\rho_1}(s_2, a_2) = 2.023$.

To verify that this Q-function is a fixed point of the contractual Bellman optimality operator (13), observe that the optimality equations hold in all state-action pairs. For example, in $(s_1, a_2)$ we have:

$$
\begin{aligned}
2.048 = q^{\rho_1}(s_1, a_2) &= \mathbb{E}_{o, s'}[-\rho_1(s_1)(o) + r^p(s_1, o) + \gamma \max_{a'} q^{\rho_1}(s', a')] = \\
&0.1(0 + 0 + 0.9 \cdot 2.048) + 0.9(-1.25 + 1.5 + 0.9 \cdot 2.023) = 2.048.
\end{aligned}
$$

Incentivizing $a_1$ in $s_2$ requires offering a contract $\rho(s_2) = (2.5, 0)$, which is not worth it for the principal, as evidenced by $q^{\rho_1}(s_2, a_1) < q^{\rho_1}(s_2, a_2)$.

**Iteration 2: agent**

The principal $\rho_1$ underestimated the payment required to incentivize effort, and the agent's best-responding policy $\pi_2$ still never invests an effort: $\pi_2(s_1, \rho_1(s_1)) = a_1$, $\pi_2(s_2, \rho_1(s_2)) = a_2$. This corresponds to the following truncated Q-function: $\overline{Q}^{\pi_2}(s_1, a_1) = 0.723$, $\overline{Q}^{\pi_2}(s_1, a_2) = -0.598$, $\overline{Q}^{\pi_2}(s_2, a_1) = -1.277$, and $\overline{Q}^{\pi_2}(s_2, a_2) = 0.402$.

To verify that this Q-function is a fixed point of the truncated Bellman optimality operator (9), observe that the optimality equations hold in all state-action pairs. For example, in $(s_1, a_2)$ we have:

$$
\begin{aligned}
-0.598 = \overline{Q}^{\pi_2}(s_1, a_2) &= -c(s_1, a_2) + \gamma \mathbb{E}_{o, s', o'}[\rho(s')(o') + \max_{a'} \overline{Q}^{\pi_1}(s', a')] = \\
&-1 + 0.9[0.1(0.9(0 + 0.723) + 0.1(1.25 + 0.402)) + \\
&\qquad 0.9(0.1(0 + 0.723) + 0.9(0 + 0.402))] = -0.598.
\end{aligned}
$$

Given the contracts from the principal's policy $\rho_1$, we can compute the agent's Q-values using (1): $Q^{\pi_2}((s_1, \rho_1(s_1)), a_1) = 0.848$, $Q^{\pi_2}((s_1, \rho_1(s_1)), a_2) = 0.527$, $Q^{\pi_2}((s_2, \rho_1(s_2)), a_1) = -1.277$, and $Q^{\pi_2}((s_2, \rho_1(s_2)), a_2) = 0.402$. Observe that indeed, the agent still prefers not to invest an effort in $s_1$, as evidenced by $Q^{\pi_2}((s_1, \rho_1(s_1)), a_1) > Q^{\pi_2}((s_1, \rho_1(s_1)), a_2)$.

**Iteration 2: principal**

The principal gives up on incentivizing effort and once again offers no contracts: $\rho_2(s_1) = (0, 0)$, $\rho_2(s_2) = (0, 0)$. This corresponds to the following contractual Q-function: $q^{\rho_2}(s_1, a_1) = 1.661$, $q^{\rho_2}(s_1, a_2) = 1.503$, $q^{\rho_2}(s_2, a_1) = 1.422$, and $q^{\rho_2}(s_2, a_2) = 1.839$.

To verify that this Q-function is a fixed point of the contractual Bellman optimality operator (13), observe that the optimality equations hold in all state-action pairs. For example, to incentivize $a_2$ in $s_1$, the principal has to offer a contract $\rho(s_1) = (0, 1.652)$, which gives us:

$$1.503 = q^{\rho_2}(s_1, a_2) = \mathbb{E}_{o,s'}[-\rho_2(s_2)(o) + r^p(s_1, o) + \gamma \max_{a'} q^{\rho_2}(s', a')] =$$

$$0.1(0 + 0 + 0.9 \cdot 1.661) + 0.9(-1.652 + 1.5 + 0.9 \cdot 1.839) = 1.503,$$

Incentivizing $a_1$ in $s_2$ requires offering a contract $\rho(s_2) = (2.098, 0)$, which is still not worth it for the principal, as evidenced by $q^{\rho_2}(s_2, a_1) < q^{\rho_2}(s_2, a_2)$.

**Subsequent iterations**

Because the principal's subgame-perfect policy $\rho_2$ repeats the policy $\rho_0$ from a previous iteration, the meta-algorithm is now stuck in a cycle where iterations 1 and 2 repeat indefinitely. In other words, the meta-algorithm diverges.

### B.7  META-ALGORITHM IN THE OBSERVED-ACTION MODEL

In the special case where the agent's actions are observed (defined in Section 2.2 and summarized in Table 2), the meta-algorithm can be shown to find SPE in a single (rather than $T + 1$) iteration, as we show in Theorem B.13. This property is based on the observation that the agent is indifferent between an MDP without a principal and an MDP augmented with a subgame-perfect principal; similar observation has been made in contemporaneous work (Ben-Porat et al., 2024). Consequently, evaluating minimal implementation only requires access to the agent's optimal Q-function in the absence of the principal.

Is is also easy to show that SPE coincides with Stackelberg equilibrium in the observed-action scenario (Lemma B.15), and consequently the meta-algorithm finds Stackelberg equilibrium.

**Theorem B.13.** *Given a principal-agent stochastic game, $\mathcal{G}$, with observed actions and either finite or infinite horizon, if the principal's policy is initialized to offer zero-vectors as contracts in all states, the meta-algorithm finds SPE in one iteration.*

*Proof.* Use the same notations of $\rho_i$ and $\pi_i$ as in the proof of Theorem 3.3 in Appendix B.4.

After initializing $\rho_0$ (that always offers $\mathbf{0}$) and finding the best-responding agent $\pi_1$, consider the optimal payments found at the outer optimization level of Algorithm 1. Given a state $s \in S$, denote the agent's action as

$$a^* = \arg\max_a Q^*((s, \mathbf{0}), a \mid \rho_0).$$

For now, let contracts in states other than $s$ remain $\mathbf{0}$; we will omit the conditioning of $Q^*$ on $\rho_0$ for brevity. The optimal contract in $s$, denoted as $b^*$, reimburses the agent for taking a suboptimal action, paying exactly

$$b^*(s, a^p) = Q^*((s, \mathbf{0}), a^*) - Q^*((s, \mathbf{0}), a^p)$$

if the agent selects $a^p$, and pays 0 otherwise. Note that $b^* = \mathbf{0}$ if $a^p = a^*$. This contract makes the agent indifferent between $a^p$ and $a^*$ because

$$Q^*((s, b^*), a^p) = Q^*((s, \mathbf{0}), a^p) + b^*(s, a^p) = Q^*((s, \mathbf{0}), a^*),$$

changing the agent's action in $s$ to $a^p$ (according to tie-breaking). However, the agent's value function remains unchanged:

$$V^*(s, b^*) = Q^*((s, b^*), a^p) = Q^*((s, \mathbf{0}), a^*) = V^*(s, \mathbf{0}).$$

Thus, the Bellman optimality equations (7) and (8) still hold in all states after replacing $\mathbf{0}$ with $b^*$ in $s$, and $\pi_1$ still best-responds to the updated principal. This replacement of zero-vectors for optimal contracts $b^*$ is performed in all states during the update of the principal at the outer optimization level, yielding a principal's policy $\rho_1$ subgame-perfect against $\pi_1$. At the same time, $\pi_1$ remains best-responding against $\rho_1$. Thus, $(\rho_1, \pi_1)$ is an SPE.  $\square$

*Remark* B.14. Unlike our general Theorem 3.3, the above proof relies on the specifics of our model such as the principal's action set and the players' reward functions.

**Lemma B.15.** *Given a principal-agent stochastic game, $\mathcal{G}$, with observed actions, any SPE is a Stackelberg equilibrium.*

*Proof.* Consider an SPE. As discussed in the above proof, the contracts in SPE exactly reimburse the agent for choosing suboptimal actions, and the agent's value function when offered such a contract in some state $s$ remains the same as in the absence of the contract. Additionally, notice that the principal may never decrease the agent's value in any state below the value it gets in the absence of contracts (since payments are non-negative). Thus, the principal may not deviate from SPE by decreasing the agent's value in any of the future states through suboptimal contracts in order to incentivize a suboptimal action $a^p$ while paying less in $s$.

On the other hand, consider the principal trying to pay less in some state $s$ by increasing the agent's value in some future states. If transition function is determenistic, then in order to decrease the payment in $s$ by some $v < b^*(a^p)$ while still incentivizing $a^p$, the principal must, for example, increase the payment in $s' = \mathcal{T}(s, \mathcal{O}(s, a^p))$ by $v/\gamma$ – which is inconsequential for the principal's value in $s$ (in our model, the discount factor is the same for the principal and the agent). In case of a stochastic transition function, the increase of payments in future states required to balance the decrease of the payment in $s$ by $v$ may even decrease the principal's value in $s$ compared to SPE.

Thus, the principal may not deviate from an SPE (commit to non-credible threats) to increase its value in the initial state, and therefore any SPE is a Stackelberg equilibrium. $\square$

## B.8    Deriving the Linear Program (Section 4)

In Section 4, we describe an RL approach to solving the principal's MDP that involves two interdependent tasks of 1) learning a policy that the principal wants to implement and 2) computing the contracts that do so optimally. The former is solved by learning the contractual Q-function (2), which we derive as a fixed point of a contraction operator $\mathcal{H}_\rho$ (13) in Appendix B.3. The latter (which is required as a subroutine to apply $\mathcal{H}_\rho$) we solve using Linear Programming, akin to how the static principal-agent problems are typically solved (as mentioned in Section 2.1).

Given a state $s \in S$ and an action to recommend $a^p \in A$, rewrite the right-hand side of $H_\rho$ as the following constrained optimization problem:

$$\max_{b \in B} \mathbb{E}_{o \sim \mathcal{O}(s, a^p), s' \sim \mathcal{T}(s, o)} \Big[ r^p(s, o) - b(o) + \gamma \max_{a'} q(s', a') \Big] \quad \text{s.t.} \tag{16}$$
$$\forall a \in A : Q^*((s, b), a^p \mid \rho) \geq Q^*((s, b), a \mid \rho),$$

where the constraints explicitly require the recommended action $a^p$ to be at least as 'good' for the agent as any other action $a$. Note that $r^p(s, o)$ and $q(s', a')$ are constants with respect to $b$ and can be omitted from the objective. To see that the constraints are linear, apply the Bellman optimality operator to the agent's Q-function, and rewrite constraints through the truncated Q-function using (1):

$$\max_{b \in B} \mathbb{E}_{o \sim \mathcal{O}(s, a^p)}[-b(o)] \quad \text{s.t.} \tag{17}$$
$$\forall a \in A : \mathbb{E}_{o \sim \mathcal{O}(s, a^p)}[b(o)] + \overline{Q}^*(s, a^p \mid \rho) \geq \mathbb{E}_{o \sim \mathcal{O}(s, a)}[b(o)] + \overline{Q}^*(s, a \mid \rho).$$

Because both the objective and the conditions are linear, the problem (17) is an LP.

As discussed in Section 4, our approach to solving the agent's optimization problem is to learn the truncated Q-function $\overline{Q}^*(\rho)$ and transform it into the Q-function by adding the expected payment. Note that this representation of the agent's Q-function is used in the constraints of the above LP. The requirement of the principal having access to the agent's (truncated) Q-function can be seen as a limitation of the outlined approach. A potential remedy is to instead parameterize the principal's Q-function $Q^*(\pi)$ with a discontinuous neural network able to efficiently approximate the Q-function and the solutions to LPs without requiring access to the agent's private information, thus extending the method of Wang et al. (2023). Of course, one could also directly learn $Q^*(\pi)$ with simpler approaches, e.g., by discretizing the contract space or employing deep RL methods for continuous action spaces (such as Deep Deterministic Policy Gradient or Soft Actor-Critic).

Solving the LP also requires access to the outcome function $\mathcal{O}$; we discuss in Appendix D.1 how this can be circumvented by parameterizing the outcome function with an additional neural network, trained as a probabilistic outcome classifier.

In the special case of the observed-action model, the LP has a simple solution if the agent best-responds to a specific principal that always offers a zero-vector $\mathbf{0}$ as a contract. In this solution, the principal exactly reimburses the agent for choosing a (suboptimal) recommended action $a^p$, and pays 0 if agent chooses any other action $a \neq a^p$. Similar observation has been made in contemporaneous work, see end of Section 3.1 in Wu et al. (2024).

### B.9 APPROXIMATION ERROR

In this section, we analyze the convergence of the meta-algorithm in presence of approximation errors. We first examine the training phase, showing how errors in principal's and agent's Q-functions propagate through iterations. During the validation phase, these errors may cause the agent to deviate from the (already suboptimal) strategy recommended by the principal, jeopardizing the principal's utility. As a countermeasure, we propose a nudging method that ensures agent's compliance through additional payments.

Recall that the meta-algorithm finds SPE by iteratively computing principal's and agent's optimal policies for $T + 1$ iterations, where $T$ is horizon. To connect our analysis to our learning-based implementation, we assume these policies are expressed through Q-functions.

#### B.9.1 TRAINING PHASE: ERROR PROPAGATION ANALYSIS

**Notation**  From Appendix B.4, recall our notation of $S_t$ for the subset of possible states of the MDP at timestep $t$, and that some time step is unique for any state ($s \in S_t$ for some $t$ and sets $S_t$ are disjoint). Recall also that $\rho_i$ and $\pi_i$ denote principal's and agent's policies found at iteration $i$ of the meta-algorithm. For convenience, let $t$ denote both the timestep and the iteration of the meta-algorithm, and iterate from $T$ to $0$ in a sequence $t \in (T, T-1, \ldots, 0)$.

Each iteration $t$, due to early stopping, both Q-functions are approximated with some errors, resulting in approximately optimal policies. Denote the approximate agent's Q-function found at iteration $t$ as $\tilde{Q}_t$; it produces a policy $\tilde{\pi}_t$ defined by $\tilde{\pi}_t(s, b) = \arg\max_a \tilde{Q}_t((s, b), a)$. Denote the approximate principal's contractual Q-function found at iteration $t$ as $\tilde{q}_t$; it produces a policy $\tilde{\rho}_t$ defined by $\tilde{\rho}_t(s) = \arg\max_a \tilde{q}_t(s, a)$.

Denote the (maximal) approximation errors of principal's and agent's Q-functions at iteration $t$ as $\delta_t$ and $\epsilon_t$ – these are defined in the next subsection. Define the accumulated approximation errors at iteration $t$ as $D_t = \max_{s \in S_t, a \in A} |\tilde{q}_t(s, a) - q^*(s, a \mid \tilde{\pi}_t)|$ and $E_t = \max_{s \in S_t, a \in A, b \in B} |\tilde{Q}_t((s, b), a) - Q^*((s, b), a \mid \tilde{\rho}_t)|$. Here, $q^*(s, a \mid \tilde{\pi}_t)$ and $Q^*((s, b), a \mid \tilde{\rho}_t)$ denote the players' Q-functions without the approximation errors (as defined in Sections 2.2 and 4): $q^*(\tilde{\pi}_t)$ is the principal's contractual Q-function when adopting the subgame-perfect policy $\rho^*(\tilde{\pi}_t)$, and $Q^*(\tilde{\rho}_t)$ is the agent's Q-function when adopting the best-responding policy $\pi^*(\tilde{\rho}_t)$. Note that these are not Q-functions in SPE.

**Backward induction in standard MDP**  To estimate $D_t$ and $E_t$, we need to analyze how approximation errors propagate through iterations. Because the meta-algorithm implicitly performs backward induction (Remark B.10), we can invoke results for backward induction in standard MDPs. Below we derive these in detail, and later derivations for principal's and agent's MDPs can be done by analogy.

Consider a standard MDP (as defined in Appendix B.2). An iteration $t$ of backward induction applies Bellman backup to the Q-function in all states in $S_t$:

$$\forall s \in S_t, a \in A : Q(s, a) = \mathbb{E}_{s' \sim \mathcal{T}(s,a)}[r(s, a, s') + \gamma \max_{a'} Q(s', a')],$$

under the condition that in terminal states $Q(s', a') \equiv 0$. Iterating until reaching the root state $s_0$ solves the MDP: the resulting policy $\pi(s) = \arg\max_a Q(s, a)$ achieves the optimal utility $u^* = V(s_0) = \max_a Q(s_0, a)$. We denote $Q^\pi \equiv Q$ to indicate that this Q-function is with respect to the policy $\pi$ – this will be useful below.

Now, consider backward induction with approximation error: the Bellman backup is applied with some error term $\varepsilon_{s,a}$ such that $|\varepsilon_{s,a}| \leq \delta$, where $\delta$ is its upper bound:

$$\forall s \in S_t, a \in A : \tilde{Q}(s,a) = \mathbb{E}_{s' \sim \mathcal{T}(s,a)}[r(s,a,s') + \gamma \max_{a'} \tilde{Q}(s',a')] + \varepsilon_{s,a},$$

where $\tilde{Q}$ denotes the resulting approximate 'Q-function' – formally, it is not a Q-function, but a numerical approximation. The policy $\tilde{\pi}(s) = \arg\max_a \tilde{Q}(s,a)$ achieves some utility $u = V^{\tilde{\pi}}(s_0) = \max_a Q^{\tilde{\pi}}(s_0, a)$ (note the superscript).

**Lemma B.16.** *In a standard MDP, if the approximation error at each iteration $t$ of backward induction does not exceed $\delta$, then the total error $\max_a |\tilde{Q}(s_0,a) - Q(s_0,a)|$ in $s_0$ is at most $D_0 = \frac{1-\gamma^{T+1}}{1-\gamma}\delta$.*

*Proof.* Each iteration, the error from previous iterations decreases by a factor of $\gamma$ due to the contraction property of the Bellman operator. Concretely, if the accumulated error in any $s' \in S_{t+1}$ is at most $D_{t+1}$, then in any $s \in S_t$ it is at most $D_t = \max_{a \in A, |\varepsilon_{s,a}| \leq \delta} |\tilde{Q}(s,a) - Q(s,a)| = \delta + \gamma \max_a |\mathbb{E}_{s'} \max_{a'}[\tilde{Q}(s',a') - Q(s',a')]| = \delta + \gamma D_{t+1}$. Solving the recursive equation with $D_T = \delta$, the accumulated error at iteration $t$ is at most $D_t = \sum_{k \in [t,T]} \gamma^{k-t}\delta$, and in particular, the total error is at most $D_0 = \frac{1-\gamma^{T+1}}{1-\gamma}\delta$ (using the sum of geometric series formula). This result shows that $\tilde{Q}(s,a)$ is within $D_t$ from $Q^{\pi}(s,a)$ in any state and for any action. $\qquad\square$

**Lemma B.17.** *In a standard MDP, if the approximation error at each iteration $t$ of backward induction does not exceed $\delta$, then the agent's utility is within $2D_0$ from the optimal, $u \geq u^* - 2D_0$.*

*Proof.* In any state $s \in S$, we have that $|Q^{\pi}(s,\pi(s)) - Q^{\tilde{\pi}}(s,\tilde{\pi}(s))| = |Q^{\pi}(s,\pi(s)) - Q^{\tilde{\pi}}(s,\tilde{\pi}(s)) + \tilde{Q}(s,\tilde{\pi}(s)) - \tilde{Q}(s,\tilde{\pi}(s))| \leq |Q^{\pi}(s,\pi(s)) - \tilde{Q}(s,\tilde{\pi}(s))| + |Q^{\tilde{\pi}}(s,\tilde{\pi}(s)) - \tilde{Q}(s,\tilde{\pi}(s))|$. For the first term, from Lemma B.16 it follows that the maximums over actions are also within $D_t$: $|Q^{\pi}(s,\pi(s)) - \tilde{Q}(s,\tilde{\pi}(s))| = |\max_a Q^{\pi}(s,a) - \max_a \tilde{Q}(s,a)| \leq D_t$. For the second term, we can recursively show (like in Lemma B.16) that: $|Q^{\tilde{\pi}}(s,\tilde{\pi}(s)) - \tilde{Q}(s,\tilde{\pi}(s))| \leq \delta + \gamma\mathbb{E}|Q^{\tilde{\pi}}(s',\tilde{\pi}(s')) - \tilde{Q}(s',\tilde{\pi}(s'))| = D_t$. Combining and substituting for $s_0$, we have that $u^* - u = Q^{\pi}(s_0,\pi(s_0)) - Q^{\tilde{\pi}}(s_0,\tilde{\pi}(s_0)) \leq 2D_0$. $\qquad\square$

**Backward induction in principal-agent MDP**  Analogous reasoning to Lemma B.16 can be applied to Principal-Agent MDPs. To simplify, we will assume that at iteration $t$, the meta-algorithm only changes Q-values in $S_t$ – akin to backward induction. The derived bounds will be loose as we ignore that by the contraction property (Theorem 3.4), each iteration improves Q-values in all states and not just in $S_t$.

At iteration $t$, the meta-algorithm first implicitly applies the Bellman backup (8) to the agent's Q-function with some noise $\varepsilon_{s,a}$ such that $|\varepsilon_{s,a}| \leq \epsilon_t$, where $\epsilon_t$ is its upper bound:

$$\tilde{Q}_t((s,b),a) = \mathbb{E}\Big[r(s,a) + b(o) + \gamma \max_{a'} \tilde{Q}_{t+1}((s',\tilde{\rho}_{t+1}(s')),a')\Big] + \varepsilon_{s,a}. \tag{18}$$

Analogously with Lemma B.16, the accumulated error at iteration $t$ is at most $E_t = \sum_{k \in [t,T]} \gamma^{k-t}\epsilon_k$, and the total error is at most $E_0 = \sum_{k \in [0,T]} \gamma^k \epsilon_k \leq \frac{1-\gamma^{T+1}}{1-\gamma}\epsilon$, where $\epsilon = \max_t \epsilon_t$.

Then, the meta-algorithm implicitly applies the Bellman backup (13) to the principal's Q-function with some noise $\varepsilon_{s,a}^p$ such that $|\varepsilon_{s,a}^p| \leq \delta_t$, where $\delta_t$ is its upper bound:

$$\tilde{q}_t(s,a) = \max_{\{b|\tilde{\pi}_t(s,b)=a\}} \mathbb{E}\Big[r^p(s,o) - b(o) + \gamma \max_{a'} \tilde{q}_{t+1}(s',a')\Big] + \varepsilon_{s,a}^p. \tag{19}$$

Analogously with Lemma B.16, the accumulated error is $D_t = \sum_{k \in [t,T]} \gamma^{k-t}\delta_k$, and the total error is $D_0 = \sum_{k \in [0,T]} \gamma^k \delta_k \leq \frac{1-\gamma^{T+1}}{1-\gamma}\delta$, where $\delta = \max_t \delta_t$.

The utility analysis is deferred until after the validation phase analysis. Note that the errors $\varepsilon_{s,a}$ and $\varepsilon_{s,a}^p$ are not dependent on the contract $b$. With this we indicate that the agent precisely estimates the expected payments of different actions when observing a contract, and the principal precisely solves the conditional maximization in $\tilde{q}_t(s,a)$.

The remaining question is what are the bounds $\epsilon_t$ and $\delta_t$. If inner and outer levels are solved with Q-learning, we can invoke known Q-learning convergence rates to tie $\epsilon_t$ and $\delta_t$ to the budget of Q-learning iterations. For example, Szepesvári (1997) proves a convergence rate of $\sqrt{\log \log(I)/I}$ (where $I$ is the number of iterations of Q-learning), so the approximation errors $\epsilon_t$ and $\delta_t$ can be chosen as proportional to this rate.

The analysis so far does not capture the interdependence of the agent's and principal's approximation errors. At the same time, this interdependence makes it difficult to analyze potential losses of principal's utility due to the errors. We address these issues below.

### B.9.2 Validation phase: utility losses and nudging

**Utility loss at validation**   Denote principal's optimal utility (given in SPE) as $u^* = V^*(s_0) = \max_a q^*(s_0, a)$. Without loss of generality, assume that if 1) the principal offers no contracts and 2) the agent minimizes the principal's utility in the MDP (the agent is adversarial), then the principal's utility equals $u = 0$. Also without loss of generality, assume that the MDP terminates at $T$ and not earlier.

The output of the training phase is an approximate principal's policy $\tilde{\rho}$. If the principal had oracle access to the agent's Q-function in SPE, then by Lemma B.17, it would be guaranteed a utility within the total approximation error, equal to $u \geq u^* - 2D_0$. However, at the training phase, the principal instead uses an estimate of the agent's Q-function.

Consider a validation phase where an oracle agent precisely (without approximation) best-responds: rather than $\tilde{\pi}$ that the principal wishes to incentivize, the oracle agent employs the best-responding policy $\pi^*(\tilde{\rho})$ by maximizing the true Q-function: $\pi^*(s, b \mid \tilde{\rho}) = \arg\max_a Q^*((s, b), a \mid \tilde{\rho})$. If these policies differ, no guarantees can be given for the principal's utility due to its discontinuity. This issue is not unique to our model but is central to principal-agent theory (see e.g. Wang et al. (2023)). In the worst case when the errors are such that the oracle agent is adversarial, the principal misses on the entirety of $u^*$ and may still pay, resulting in a negative utility, $u < 0$. Below we derive a method for improving this worst-case utility guarantee.

**Nudging**   The only way to mitigate this risk is through additional payments, which we call 'nudging' the agent. Nudging is intended to compensate the potential errors in the offered contracts that would cause the oracle agent to deviate. Concretely, without nudging, the incentive-compatibility constraints in the LP (3) only require that the agent weakly prefers some action, whereas with nudging, the agent must strictly prefer it by some margin of $\xi_t$ (this margin is added to the right-hand side of LP constraints). Below we describe an implementation of nudging that improves the utility lower bound.

The nudges must be added when finding optimal contracts during training, as they change the players' Q-functions. Clearly, this changes the solution that the meta-algorithm finds. Given values of $\xi_t$, we term this modified solution concept as $\xi$-SPE. Analogously to how $\tilde{Q}$ and $\tilde{q}$ denote approximations of agent's and principal's Q-functions in SPE, denote these approximations for $\xi$-SPE as $\tilde{Q}^\xi$ and $\tilde{q}^\xi$. Applying meta-algorithm with error to approximate $\xi$-SPE, the Q-function approximations are updated similarly to (18) and (19):

$$\tilde{Q}_t^\xi((s, b), a) = \mathbb{E}\Big[r(s, a) + b(o) + \gamma \max_{a'} \tilde{Q}_{t+1}^\xi((s', \tilde{\rho}_{t+1}^\xi(s')), a')\Big] + \varepsilon_{s,a}, \tag{20}$$

$$\tilde{q}_t^\xi(s, a) = \max_{\{b \mid \tilde{\pi}_t(s,b) = a\}} \mathbb{E}\Big[r^p(s, o) - b(o) + \gamma \max_{a'} \tilde{q}_{t+1}^\xi(s', a')\Big] + \varepsilon_{s,a}^p, \tag{21}$$

At the training phase, computing optimal contracts involves solving LPs with approximate Q-functions (explicitly or otherwise). Consider such LPs in the problem with nudging. For a state $s \in S_t$ and a desirable action $a^p \in A$, the contract is computed as a solution to:

$$\min_{b \in B} \mathbb{E}[b(o)] \quad \text{s.t.} \quad \forall a \in A : \tilde{Q}_t^\xi((s, b), a^p) \geq \tilde{Q}_t^\xi((s, b), a) + \xi_t. \tag{22}$$

By design, $\tilde{Q}_t^\xi((s, b), a)$ is within $E_t$ from $Q^*((s, b), a \mid \tilde{\rho}_t^\xi)$ (and the value of $E_t$ is derived in Appendix B.9.1). So, to guarantee the oracle agent's choice of $a^p$ in $s$, i.e. that $\forall a : Q^*((s, b), a^p \mid$

$\tilde{\rho}_t^\xi) \geq Q^*((s, b), a \mid \tilde{\rho}_t^\xi)$, it is sufficient to choose $\xi_t = 2E_t$ in (22). This choice of nudges ensures the oracle agent's compliance across all states.

Denote the contract that solves the LP (22) for $\xi_t = 0$ (so, without nudging in $s$) as $b^*$, and the contract that solves it for $\xi_t = 2E_t$ as $b^\xi$. Denote the increase in the expected payment resulting from a nudge of $\xi_t$ as $\sigma_{s,a^p} = \mathbb{E}[b^\xi(o) - b^*(o)]$. Denote the optimal principal's utility in $\xi$-SPE as $u^\xi$. By design, the difference between optimal utilities is at most $u^* - u^\xi \geq \sum_t \gamma^t \mathbb{E}_{s \in S_t} \sigma_{s,a^p}$, where the expectation over states is with respect to the probability of the MDP being in state $s$ at iteration $t$ under the principal's and agent's policies (and $a^p$ maximizes principal's Q-function in $s$).

Notice that if the constraint is tight without the nudge, then the increase of expected payment is at least $\sigma_{s,a^p} \geq \xi_t$. The equality holds, for example, in the observed-action model: the optimal contract $b^*$ directly offers a payment for performing a certain action, and a nudge of $\xi_t = 2E_t$ simply increases this payment by as much. In this case, a guaranteed utility is $u \geq u^\xi - 2D_0 \geq u^* - 2D_0 - 2\sum_{t \in [0,T]} \gamma^t E_t$.

For the general case, $\sigma_{s,a^p}$ needs to be bounded from above. This bound will depend on how much the outcome distributions between $a^p$ and other actions overlap. Specifically, assuming there are at least as much outcomes as actions, $|A| \leq |O|$, then for any action $a^p$, there is always an outcome $o^p$ with higher mass for this action than for other actions: $\forall s \in S, a^p \in A, \exists o^p \in O : \forall a \in A, \mathcal{O}(s, a^p)(o^p) > \mathcal{O}(s, a)(o^p)$. Therefore, given $s$ and $a^p$, increasing the payment along this outcome by $\frac{\xi_t}{\mathcal{O}(s,a^p)(o^p) \cdot \min_a[\mathcal{O}(s,a^p)(o^p) - \mathcal{O}(s,a)(o^p)]}$ leads to an increase of the expected payment by $\sigma_{s,a^p} = \frac{\xi_t}{\min_a[\mathcal{O}(s,a^p)(o^p) - \mathcal{O}(s,a)(o^p)]}$ and guarantees that the nudged constraints are satisfied. Denoting the minimal value of the denominator across all states and actions as

$$d_{\min} = \min_{s \in S, a \in A, a^p \in A} [\mathcal{O}(s, a^p)(o^p) - \mathcal{O}(s, a)(o^p)], \tag{23}$$

the guaranteed utility is $u \geq u^* - 2D_0 - 2\sum_{t \in [0,T]} \gamma^t E_t / d_{\min}$.

## C  PRINCIPAL-MULTI-AGENT EXAMPLE: PRISONER'S DILEMMA

Below we illustrate the multi-agent model from Section 5.1 on the example of a simple matrix game.

Consider a standard one-step Prisoner's Dilemma with the payoff matrix as in Table 3a. Here, the only equilibrium is mutual defection (DD), despite cooperation (CC) being mutually beneficial. How should a benevolent principal change the matrix through payments to incentivize cooperation?

One answer is that CC should become an equilibrium through minimal payments in CC. In one of the payment schemes that achieve this, the principal pays a unit of utility to both players for the CC outcome, resulting in the payoff matrix as in Table 3b.[4] However, note that DD remains an equilibrium for the agents. In fact, the new payoff matrix is also a social dilemma known as the Stag Hunt. In the context of (decentralized) learning, there is no reason to expect the agents to converge to CC instead of DD, and convergence to suboptimal equilibria has been observed empirically in generalized Stag Hunt games (Peysakhovich & Lerer, 2018b).

As a more robust approach, the principal can make action C dominant. This is stronger than just making CC an equilibrium because each agent will prefer action C regardless of the actions of others. To achieve this, in addition to paying a unit of utility to both agents in CC, the principal pays two units of utility to the cooperating agent in CD and DC, resulting in the payoff matrix as in Table 3c. This is the kind of solution we implement in our application to SSDs, where we formulate the principal's objective as learning a social welfare maximizing strategy profile and its minimal implementation.

Importantly, in the context of our principal-agent model, the minimal implementation is consistent with SPE in that the principal minimizes payments when agents best-respond by following recommendations (CC in our example). So the IC property is an additional constraint, which specifies an otherwise ambiguous objective of finding the principal's policy in SPE, and which only requires additional payments in equilibrium.

---

[4]Unlike the single-agent model, the optimal payment scheme here is ambiguous. Any payment scheme that gives a unit of utility to both agents in CC and no utility to a defecting agent in DC and CD forms an SPE when coupled with best-responding agents: CC becomes an equilibrium, the payments in which are minimized.

Table 3: Prisoner's Dilemma as a principal-multi-agent game. 'Def' denotes 'Defect' and 'Coop' denotes 'Cooperate'. Blue highlights principal's changes of agents' payoffs.

| (a) no payments | | | (b) arbitrary payments in SPE | | | (c) IC payments in SPE | | |
|---|---|---|---|---|---|---|---|---|
| | Def | Coop | | Def | Coop | | Def | Coop |
| Def | 2, 2 | 4, 0 | Def | 2, 2 | 4, 0 | Def | 2, 2 | 4, 2 |
| Coop | 0, 4 | 3, 3 | Coop | 0, 4 | 4, 4 | Coop | 2, 4 | 4, 4 |

Note that in our example, the payment to each agent for the same action depends on the other agent's action (e.g., the row agent receives +2 in CD and +1 in CC). This dependence on other agents is even more pronounced in stochastic games, where payments of a minimal implementation condition on the *policies* of others – not only on their immediate actions in the current state but also on their actions in all possible future states.[5] For this reason, learning the precise minimal implementation is generally impractical: consider a neural network used for contract estimation for some agent explicitly conditioning on the parameters of neural networks of all other agents, or on their actions in all states of the MDP.

As a tractable alternative, we use a simple approximation that performs well in our experiments. Specifically, we train the principal assuming that each agent sticks to one of two policies: either always follow the principal's recommendations and get paid, or ignore them and act independently as if there is no principal. In equilibrium, both policies maximize the agent's welfare, which justifies the assumption. For implementation details of this approximation, see Appendix D.2.

# D EXPERIMENTS

## D.1 EXPERIMENTS IN TREE MDPS

In this section, we empirically test the convergence of Algorithm 1 towards SPE in hidden-action MDPs. We experiment with a small variation on the algorithm, which expedites convergence: the agent and principal policies are updated simultaneously instead of iteratively. This can be seen as terminating inner and outer tasks early, after one update, leading to approximately optimal policies.

**Environment.** In these experiments, we simulate a *multi-stage project* in which the principal offers contracts at intermediate stages, influencing the agent's choice of effort. Specifically, we model an MDP that is a complete binary tree, where the agent's decisions at each state are binary, representing high or low effort, and correspond to good or bad outcomes with different probabilities. This is an intentionally simple environment, allowing us to compare with precise ground truth.

The MDP is represented by a complete binary decision tree with depth 10, resulting in 1023 states. In each state, the agent may take two actions, $a_0$ and $a_1$, which may result in two outcomes, $o_0$ and $o_1$. The action $a_0$ results in the outcome $o_0$ with probability 0.9 in all states; likewise, $a_1$ results in $o_1$ with probability 0.9. The tree transitions to the left subtree after outcome $o_0$ and to the right subtree after outcome $o_1$.

The action $a_0$ yields no cost for the agent, i.e., $\forall s : r(s, a_0) = 0$; and the outcome $o_0$ yields no reward for the principal, i.e. $\forall s : r^p(s, o_0) = 0$. Conversely, the action $a_1$ is always costly and the outcome $o_1$ is always rewarding, with values randomly sampled. Specifically, let $\mathbb{U}$ denote one-dimensional uniform distribution, $\mathbb{U}_1 = \mathbb{U}[0, 1]$ and $\mathbb{U}_2 = \mathbb{U}[0, 2]$. Then, the agent's reward is generated as $r(s, a_1) = -(u \sim \mathbb{U}[0, 1 - (v \sim \mathbb{U}_1)])$, and the principal's reward is generated as $r^p(s, o_1) = (u \sim \mathbb{U}[0, 2 - (v \sim \mathbb{U}_2)])$ Note that the principal's reward is on average higher than the agent's cost. Using this reward function sampling method, the principal's policy $\rho^*$ in SPE offers non-trivial contracts that incentivize $a_1$ in about 60% of states.

**Implementation details.** We parameterize the principal's and the agent's Q-functions as *Deep Q-Networks* (Mnih et al., 2015) respectively denoted by $\theta$ and $\phi$. The input to both networks is a state $s$, and both networks approximate Q-values for all actions $a \in A$. Specifically, the principal's

---

[5]Here we refer to the dependence of value functions on $\pi_{-i}$ in the IC constraints (5).

---

**Algorithm 2** A deep Q-learning implementation of Algorithm 1 in the single-agent setting

---

**Require:** principal's Q-network $\theta$, agent's Q-network $\phi$, target networks $\theta'$ and $\phi'$, replay buffer $RB$

1: Initialize buffer $RB$ with random transitions, networks $\theta$ and $\phi$ with random parameters
2: **for** `number of updates` **do**                                                                ▷ Training loop
3:   **for** `number of interactions per update` **do**                        ▷ Interact with the MDP
4:     Select an action to recommend, $a^p$, with $q^\theta(s, a)$ via $\epsilon$-greedy
5:     Sample $o \sim \mathcal{O}(s, a)$, $r^a = r(s, a)$, $r^p = r^p(s, o)$, transition the MDP to $s' \sim \mathcal{T}(s, o)$
6:     Add the transition $(s, a^p, r^a, r^p, o, d, s')$ to the buffer $RB$
7:     If $d = 1$, reset the MDP                   ▷ $d$ is a binary 'done' variable indicating termination
8:   Sample a mini-batch of transitions $mb \sim RB$
9:   **for** $(s, a^p, r^a, r^p, o, d, s') \in mb$ **do**            ▷ Estimate target variables to update $\theta$ and $\phi$
10:     $a^{p\prime} = \arg\max_{a'} q^\theta(s', a')$                    ▷ Select the next action for Q-learning updates
11:     Find optimal contracts $b^*(s, a^p)$ and $b^*(s', a^{p\prime})$ by solving LP (17)
12:     $y^p(s, a^p) = r^p - b^*(s, a^p, o) + \gamma(1 - d)q^{\theta'}(s', a^{p\prime})$
13:     $y^a(s, a^p) = r^a + \gamma(1 - d)(\mathbb{E}_{o' \sim \mathcal{O}(s', a^{p\prime})}b^*(s', a^{p\prime}, o') + \overline{Q}^{\phi'}(s', a^{p\prime}))$
14:   Minimize $L(\theta) = \sum_{mb} \left(q^\theta(s, a^p) - y^p(s, a^p)\right)^2$                    ▷ Update $\theta$ as DQN
15:   Minimize $L(\phi) = \sum_{mb} \left(\overline{Q}^\phi(s, a^p) - y^a(s, a^p)\right)^2$                    ▷ Update $\phi$ as DQN

---

network estimates the contractual optimal Q-values $q^\theta(s, a)$, representing its payoffs when optimally incentivizing the agent to take action $a$ in state $s$; and the agent's network estimates the truncated optimal Q-values $\overline{Q}^\phi(s, a)$, representing its payoffs minus the expected immediate payment.

Algorithm 2 describes our Deep Q-learning-based implementation of Algorithm 1 for the single-agent MDPs. The overall pipeline is standard, with a few notable details.

First, unlike the iterative convergence of the principal and the agent in Algorithm 1, the two policies are trained simultaneously.

Second, the outcome function $\mathcal{O}$ is assumed to be known. If not, it can be parameterized as an additional neural network $\xi$ and trained as a probabilistic classifier with sampled outcomes used as ground truth; the resulting loss function would be $L(\xi) = \sum_{mb} CE[\mathcal{O}_\xi(s, a^p), o]$, where $CE$ denotes cross-entropy, $mb$ denotes a mini-batch, and $\mathcal{O}_\xi(s, a^p)$ denotes the predicted probabilities of outcomes as a function of state-action. We implemented this in our single-agent experiments with $\mathcal{O}$ constant across states and found no difference from having access to $\mathcal{O}$, although this might change in more complex scenarios.

Third, when updating the agent's network $\phi$, the target variable $y^{\phi'}(s, a^p)$ is estimated based on the contract in the next state $s'$ rather than the current state $s$. Since payment is a part of reward, the target variable is effectively estimated as a mixture of parts of rewards in $s$ and $s'$. This is required so that the truncated rather than the standard Q-function is learned.

**Hyperparameters.**   The neural networks have 2 hidden fully connected layers, each consisting of 256 neurons and followed by a ReLU activation. The networks are trained for 20000 iterations, each iteration including 8 environment interactions and a single gradient update on a mini-batch with 128 transitions, sampled from the replay buffer. Every 100 iterations, the target networks are updated by copying the parameters of the online networks. The learning rate is initialized at 0.001 and is exponentially annealed to 0.0001 throughout the training. Similarly, the exploration rate $\epsilon$ is initialized at 1 and is linearly annealed to 0 throughout the training. Rewards are not discounted, i.e., $\gamma = 1$.

**Results.**   We generate three instances of the Tree MDP, each with randomly sampled reward functions, and run five trials on each Tree MDP. For each trial, we employ our two-phase setup from Section 4. The training phase is performed with a deep Q-learning implementation of the meta-algorithm, Algorithm 2. Rather than after the training phase, we perform validation with the oracle agent (that precisely best-responds) throughout the training, and report the players' utilities

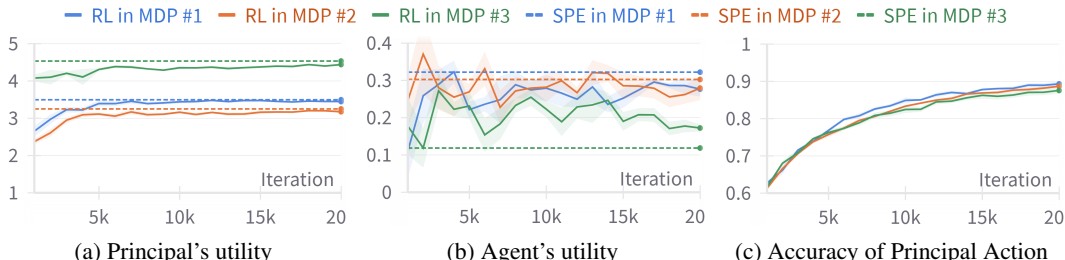

| (a) Principal's utility | (b) Agent's utility | (c) Accuracy of Principal Action |

Figure 3: Results in Tree MDPs. Solid lines are learning curves of DQNs trained with Algorithm 2; the utilities are computed by coupling the learned principal's policy with a best-responding oracle agent throughout the training. Dashed lines represent optimal utilities in SPE obtained with dynamic programming. Different colors represent three distinct instances of the tree environment. For each, we use five trials of the algorithm (shaded regions represent standard errors).

in Figure 3a and 3b. We also compare the policy that the principal incentivizes with the optimal in Figure 3c.

From Figure 3a, we observe that our algorithm attains a principal's utility of just $2\%$ below the optimal utility in SPE. On the other hand, Figure 3b shows that the agent's utility can exhibit similar (in absolute terms) deviations from SPE in either direction. The 'accuracy' metric in Figure 3c indicates that the principal recommends the action prescribed by the optimal policy in approximately $90\%$ of the states.

The small underperformance in the principal's utility and the biases in the agent's utility can be partially attributed to the $10\%$ of the non-optimal principal's actions. That around $90\%$ of correct actions amount to around $98\%$ of the principal's utility suggests errors likely occur in rarely visited states or states where actions result in similar payoffs. The minor discrepancies in utility, coupled with the learning dynamics, underscore the complex interplay between the principal and the agent as they adapt to each other throughout training.

### D.2    EXPERIMENTS IN THE COIN GAME

**Implementation details.**    Algorithm 3 describes our Deep Q-learning-based implementation of Algorithm 1 for the multi-agent MDPs with self-interested agents, often referred to as "sequential social dilemmas". The experimental procedure consists of two phases: training and validation.

In the training phase, for each agent $i$, the principal's objective is to find a policy to recommend by learning the contractual Q-function, $q_i^\theta$, as well as its minimal implementation by learning the agent's Q-function in the absence of the principal, $Q_i^\phi$. In Section 5.1, we additionally require the minimal implementation to be in dominant strategies. Learning such an implementation requires conditioning $Q_i^\phi$ on the inter-agent policies $\boldsymbol{\pi}_{-i}$, which is intractable. As a tractable approximation, given that rational agents will only deviate from recommendations if their expected payoffs are not compromised, we simplify each agent's strategy space to two primary strategies: cooperation (following the principal's recommendations) or rational deviation (default optimal policy disregarding contracts, which gives the same utility). We encapsulate this behavior in a binary variable $f_i \in \{0, 1\}$, indicating whether agent $i$ follows the recommendation $a^p$. Then, $Q_i^\phi$ is conditioned on the joint $\mathbf{f}_{-i}$ variable of the other agents, indicating both the immediate outcome and the agents' future behavior. The efficacy of this simplification is validated through our experimental results.

During the training phase, we randomly sample $f_i$ for each agent at the beginning of an episode. For the episode's remainder, each agent either follows the recommendations given by $q_i^\theta$ (if $f_i = 1$) or acts selfishly according to $Q_i^\phi$ (if $f_i = 0$). The experience generated this way is then used to train both $\theta$ and $\phi$, enhancing exploration and covering the whole space of $\mathbf{f}$.

Given the principal obtained in the training phase, the subsequent validation phase independently trains selfish agents parameterized by $\psi_i$ from scratch in the modified environment. Specifically, each agent observes an action recommendation when computing Q-values, $Q_i^{\psi_i}((s, a_i^p), a_i)$, and is paid

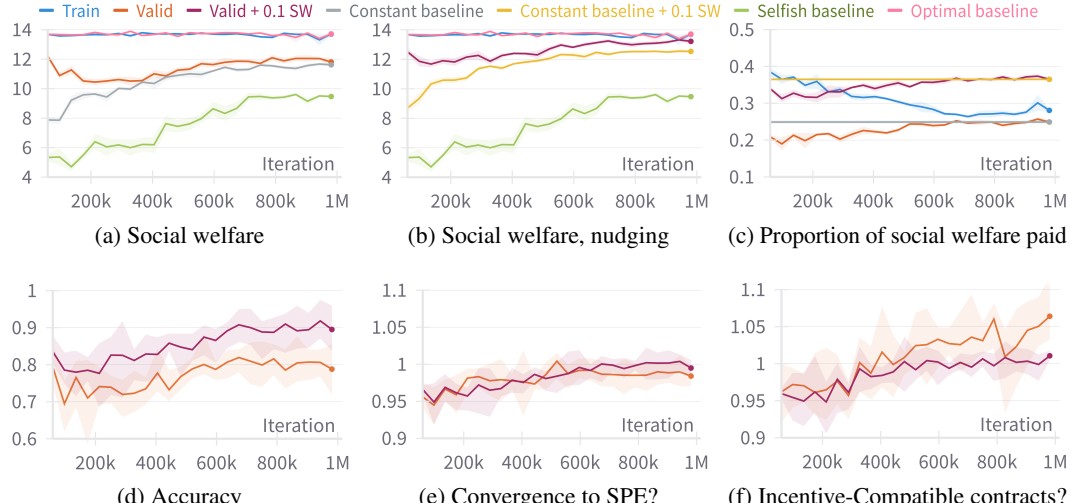

Figure 4: Learning curves in the Coin Game with a $3 \times 3$ grid and each episode lasting for $20$ time steps. The structure is the same as in Figure 2, with the addition of the top middle plot. The definition of the plots is as follows: a) total return of the two agents (without payments); b) same, but our algorithm and constant baseline additionally pay $10\%$ of social welfare to the agents; c) a ratio of the total payment by the principal to what the agents would effectively be paid if directly maximizing social welfare; d) the proportion of the principal's recommendations followed by the validation agents; r) a ratio between the utilities of an agent's policy at a given iteration and the recommended policy, with the opponent using the recommended policy; f) same ratio, but with the opponent's policy at a given iteration. Each experiment is repeated 5 times, and each measurement is averaged over 80 episodes.

by the principal if the recommendation is followed. When estimating payments, $f_i$ simply indicates whether agent $i$ followed the recommendation. Other than these changes, we use the standard DQN training procedure. We also do not assume the principal's access to the agents' private information like Q-values or parameters.

**Hyperparameters.** The neural networks consist of 1 hidden convolutional layer followed by 2 hidden fully connected layers and an output layer. The convolutions have $4 \times 4$ kernels and transform the initial $4$-channel states into 32 channels. The fully connected layers consist of $64$ neurons. All hidden layers are followed by ReLU activations. The networks are trained for $1000000$ iterations, each iteration including a single environment interaction and a single gradient update on a mini-batch with 128 transitions, sampled from the replay buffer. Every 100 iterations, the target networks are updated by copying the parameters of the online networks. The learning rate is initialized at $0.0005$ and is exponentially annealed to $0.0001$ throughout the training. Similarly, the exploration rate $\epsilon$ is initialized at $0.4$ and is linearly annealed to $0$ throughout the training. The discounting factor is set at $\gamma = 0.99$. For more efficient training, we use prioritized replay buffers (Schaul et al., 2015) with a maximum size of 100000, $\alpha = 0.4$, $\beta = 0$, and $\epsilon = 10^{-7}$.

**Additional results.** In Figure 4, we report results in the Coin Game with a $3 \times 3$ grid and each episode lasting for 20 time steps. This is a simpler environment than the one in the main text, as the state space and the horizon are smaller.

During training, our algorithm finds an approximately optimal joint policy (Fig. 4a) and estimates that about $30\%$ of social welfare is sufficient for an IC implementation (Fig. 4c). Interestingly and contrary to our previous results, the validation phase does not confirm this: we observe that the validation agents fall short of the optimal performance, as well as that our algorithm does not outperform the constant proportion baseline (Fig. 4a). On the one hand, we do not find evidence that it does not converge to SPE, as in Figure 4e, the utility ratio hovers around $1$. On the other hand, a test on incentive compatibility (Fig. 4f) reveals that the validation agents consistently find policies

that perform $5\%$ to $10\%$ better against the opponent DQNs, meaning that the principal fails to learn IC contracts. For more information on these tests, see the discussion in Section 5.3 We conjecture that this negative result is due to using the approximation of IC through $f_i$ variables during training, as described in the implementation details.

As a remedy, we implement a form of nudging intended to counteract the approximation errors, as discussed in Appendix B.9. Specifically, we increase the offered payments in each state by $10\%$ of the social welfare; we increase the proportion of the constant baseline accordingly. The effect of this modification is illustrated in Figure 4b: the performance of our algorithm reaches that of the optimal baseline, whereas the constant baseline still falls short. Furthermore, the IC property appears fixed as the utility ratio decreases to around 1 (Fig. 4f). The accuracy metric also improves, raising the frequency of agents following the principal's recommendation from around $80\%$ to around $90\%$.

Overall, we believe these results to be positive: our algorithm falls short somewhat predictably given the practical approximation of IC (and the tie-breaking assumption), and still manages to outperform the constant baseline while paying the same. As a side note, these results also showcase the usefulness of our two-step experimental procedure, as opposed to the usual performance comparison during the training phase, which does not reveal the hidden issues.

## D.3 COMPUTE

All experiments were run on a desktop PC with 16 GB RAM, 11th Gen Intel(R) Core i5-11600KF @3.90GHz processor, and NVIDIA GeForce RTX 2060 GPU. The single-agent experiments were run using only CPU, and the multi-agent experiments were run using GPU to store and train neural networks and CPU for everything else. Each run (one algorithm, one random trial) takes thirty minutes to an hour. The open-source code packages we use are PyTorch (BSD-style license) (Paszke et al., 2019), RLlib (Apache license) (Liang et al., 2018), Stable-Baselines3 (MIT license) (Raffin et al., 2021), Gymnasium (MIT license) (Towers et al., 2023), and W&B (MIT license) (Biewald, 2020).

**Algorithm 3** A deep Q-learning implementation of Algorithm 1 in the multi-agent setting

    **TRAINING PHASE**

**Require:** principal's Q-network $\theta$, agents' Q-network $\phi$, target networks $\theta'$, $\phi'$, replay buffer $RB$

1: Initialize buffer $RB$ with random transitions, networks $\theta$ and $\phi$ with random parameters
2: Sample a binary vector $\mathbf{f} = (f_i)_{i \in N}$    $\triangleright$ $f_i$ indicates if $i$ follows recommendations this episode
3: **for** number of updates **do**                         $\triangleright$ Training loop
4:     **for** number of interactions per update **do**       $\triangleright$ Interact with the MDP
5:         **for** $i \in N : f_i = 1$ **do**:             $\triangleright$ The principal acts for these agents
6:             Select a recommended action $a_i^p$ with $q_i^\theta(s, a_i^p)$ via $\epsilon$-greedy, set $a_i = a_i^p$
7:         **for** $i \in N : f_i = 0$ **do**:            $\triangleright$ These agents act selfishly ignoring payments
8:             Select an agent's action $a_i$ with $Q_i^\phi((s, \mathbf{f}_{-i}), a_i)$ via $\epsilon$-greedy
9:         **for** $i \in N$ **do**:                  $\triangleright$ Get rewards from the environment
10:             $r_i = r_i(s, \mathbf{a})$
11:         Transition the MDP to $s' \sim \mathcal{T}(s, \mathbf{a})$
12:         Add the transition $(s, \mathbf{a}, \mathbf{f}, \mathbf{r}, s')$ to the buffer $RB$
13:         If $d = 1$, reset the MDP and resample $\mathbf{f}$
14:     Sample a mini-batch of transitions $mb \sim RB$
15:     **for** $(s, \mathbf{a}, \mathbf{f}, \mathbf{r}, s') \in mb$ **do**           $\triangleright$ Estimate target variables to update $\theta$ and $\phi$
16:         **for** $i \in N$ **do**
17:             $b_i^*(s, a_i) = \max_a Q_i^\phi((s, \mathbf{1}), a) - Q_i^\phi((s, \mathbf{1}), a_i)$      $\triangleright$ As if $\mathbf{a}$ were recommended
18:             $y_i^p(s, a_i) = r_i - \alpha b_i^*(s, a_i) + \gamma \max_{a_i'} q_i^{\theta'}(s', a_i')$        $\triangleright$ we set $\alpha = 0.1$
19:             $y_i^a(s, a_i) = r_i + \gamma \max_{a_i'} Q^{\phi'}((s', \mathbf{f}_{-i}), a_i')$
20:     Minimize $L(\theta) = \sum_{mb} \left( \sum_i q_i^\theta(s, a_i) - \sum_i y_i^p(s, a_i) \right)^2$         $\triangleright$ Update $\theta$ as VDN
21:     Minimize $L(\phi) = \sum_{mb} \sum_i \left( Q_i^\phi((s, \mathbf{f}_{-i}), a_i) - y_i^a(s, a_i) \right)^2$      $\triangleright$ Update $\phi$ as DQN

    **VALIDATION PHASE**

**Require:** validation agents' Q-networks $(\psi_i)_{i \in N}$, target networks $(\psi_i')$, replay buffer $RB$

22: Initialize buffer $RB$ with random transitions, networks $\psi_i$ with random parameters
23: **for** number of updates **do**                       $\triangleright$ Training loop
24:     **for** number of interactions per update **do**       $\triangleright$ Interact with the MDP
25:         **for** $i \in N$ **do**                  $\triangleright$ Agents act selfishly
26:             $a_i^p = \arg\max_a q_i^\theta(s, a)$           $\triangleright$ Recommended action by $\theta$
27:             Select an agent's action $a_i$ with $Q_i^{\psi_i}((s, a_i^p), a_i)$ via epsilon-greedy
28:             Set $f_i = 1$ if $a_i = a_i^p$ else $f_i = 0$    $\triangleright$ $f_i$ indicates if $i$ followed recommendation in $s$
29:         **for** $i \in N$ **do**:                  $\triangleright$ Get total rewards
30:             $b_i^*(s, a_i^p) = \max_a Q_i^\phi((s, \mathbf{f}_{-i}), a) - Q_i^\phi((s, \mathbf{f}_{-i}), a_i^p)$
31:             $R_i = r_i(s, \mathbf{a}) + f_i b_i^*(s, a_i^p)$           $\triangleright$ Agent $i$ is only paid if $f_i = 1$
32:         Transition the MDP to $s' \sim \mathcal{T}(s, \mathbf{a})$
33:         Add the transition $(s, \mathbf{a}, \mathbf{R}, s')$ to the buffer $RB$
34:         If $d = 1$, reset the MDP
35:     Sample a mini-batch of transitions $mb \sim RB$
36:     **for** $i \in N$ **do**                  $\triangleright$ Independently update each Q-network
37:         **for** $(s, a_i, R_i, s') \in mb$ **do**        $\triangleright$ Estimate target variables to update $\psi_i$
38:             $a_i^{p\prime} = \arg\max_{a'} q_i^\theta(s', a')$
39:             $y_i(s, a_i) = R_i + \gamma \max_{a_i'} Q_i^{\psi_i'}((s', a_i^{p\prime}), a_i')$
40:         Minimize $L(\psi_i) = \sum_{mb} \left( Q_i^{\psi_i}((s, a_i^p), a_i) - y_i(s, a_i) \right)^2$      $\triangleright$ Update $\psi_i$ as DQN

