# OpenReview forum: "Principal-Agent Reinforcement Learning: Orchestrating AI Agents with Contracts"
_ICLR.cc/2025/Conference — Submitted to ICLR 2025_

### Official Review · Reviewer_nJAV · 2024-10-25

**Soundness:** 3
**Presentation:** 3
**Contribution:** 3
**Rating:** 6
**Confidence:** 3

**Summary:**

This paper formulates a general framework of principal-agent Reinforcement Learning (RL), which tries to train multi-agent RL agents without centralized control, but instead by using a protocol where a principal agent tries to guide (intentionally misaligned) agent behavior by assigning payments. The paper first constructs a hidden-action principal-agent stochastic game, then propose an meta-algorithm for finding Subgame Perfect Equilibrium (SPE). After analyzing the theoretical property of such meta-algorithm, the paper discusses the possibility of learning it with RL, and then finally how to extend to multi-agent RL. Several experiments show that the proposed method works well in the coin game and tree MDPs

**Strengths:**

1. This paper addresses a very interesting problem as it considers a novel scenario where the principal needs to regulate agents' behavior but does not have centralized control on each agent. This is even more interesting considering the potential future use of regulating a herd of LLM agents with a learned principal, many of which can self-reflect and improve according to a protocol but we do not have training access to.

2. This paper is well-written and well-structured. It builds up a framework with many complicated notions, but the paper managed to consider one component at a time: first it is the game itself, then an idealized solution, then function approximator (RL), and finally multi-agent. Generally, the paper is easy to follow.

3. The paper gives detailed analysis and proof for its theoretical performance (e.g., convergence property), which is important for a new, economic-inspired RL framework. In the mean time, the main text is kept clean without too many unnecessary notations using intuitive explanations.

**Weaknesses:**

1. The hyperparameter, $\alpha$ in line 406, could be tricky. As stated by the authors, $\alpha$ is set as a heuristic approach to downweight the importance of payment minimization compared to social welfare (payment minimization cannot hurt social welfare), and thus $\alpha$ should be set very small. However, if $\alpha$ is indeed very close to $0$, the principal agent can simply use its reward to overwhelm all selfish intentions (e.g. sent 1% of its own reward to each individual agent, which is a very large amount for them) and the algorithm would degenerate to independent RL with cooperative objective. Thus, there exists some balance of $\alpha$ that may require tuning. It would be better if the authors can conduct an ablation study on $\alpha$ to show how it impacts the balance between social welfare and payment minimization.

2. There is essentially only one gridworld experiment for the proposed algorithm in multi-agent RL (tree MDP in the appendix is for single-agent). It would be better if more and harder environments could also be tested, such as particle world ones (environments like SMAC-like ones would be even better for camera ready (if accepted) / next submission (if rejected)).

**Minor issues**

1. The notations in the beginning of Sec. 2.2 are unclear. For example, $o$ is an outcome, but is then used as a number ("o-th coordinate of a contract"); expressions like "contract corresponds to the outcome $o$" would be better.

2. The punctuation "." at the end of the caption of Tab. 1 is missing.

3. The current contribution part is not stating the paper's contribution, but is rather an outline of the paper. It would be better if the outlines can be more briefly summarized into a paragraph, with each highlight on key novel aspect as one sentence.

**Questions:**

I have a question: can the solution proposed by this framework improve social welfare on a more complicated, closer-to-mainstream MARL environment (e.g. a continuous version of coin game in a multi-agent particle world style)?

---

> ### Author Response · Authors · 2024-11-27
>
> We cordially thank the reviewer for these great suggestions!
>
> We agree that exploring the sensitivity to the hyperparameter alpha (e.g., via an ablation study) and additional experiments would strengthen the paper. Regarding the latter, the other reviewers also make the case that it would be interesting to add additional experiments on more collaborative tasks (on which we should expect good/better results), in addition to focusing on very challenging settings in which the agents have very conflicting objectives (although more of these would of course be good, too).
>
> We will work on both.
>
> Thanks for pointing out the minor issues. We will fix these!

---

> > ### Comment · Reviewer_nJAV · 2024-12-01
> >
> > Thanks for your reply, though I did not see the updates reported (please feel free to point out if I overlooked pdf update).

---

### Official Review · Reviewer_b1yZ · 2024-11-02

**Soundness:** 3
**Presentation:** 3
**Contribution:** 2
**Rating:** 5
**Confidence:** 4

**Summary:**

This work introduces a specific contract mechanism into MARL, defined a principal-agent stochastic game based on this mechanism, and solved the game using a straightforward value-based RL algorithm, conducting some experiments as well.

**Strengths:**

- Introducing contracts into MARL is an issue of interest to the community.
- This work is comprehensive, thoroughly addressing the bilevel MARL problem related to the contract mechanism.

**Weaknesses:**

- Algorithm 1 is vanilla, and lines 3 and 4 imply solving an MDP, which can be very time-consuming in a larger environment. Although a value-based method is used and is model-free, it is a primitive algorithm. Some analyses are straightforward to the community. The main contribution lies in introducing the contract mechanism and reformulating the game.

- Is the contract in this work a mechanism, or is it a problem to be solved?
    - As a problem: The contract reformulates a new game, and the logic of the experimental design is to verify that their proposed algorithm can solve this problem. So, could some other existing algorithms potentially solve this problem?
    - As a mechanism: For instance, with this contract mechanism, the prisoner’s dilemma can be effectively resolved, and the coin game can achieve high scores. In this sense, this work should compare with other algorithms to demonstrate the advantages of the contract. The experiments in Figure 2 are clearly insufficient.

- The contract has a very specific meaning here, where after an agent takes an action and a result is achieved, the principal pays the agent based on the result. This setting is a variant of incentivizing agents.
    - In this way, this setting is closely related to LIO ("learning to incentivize other learning agents"). The authors cited LIO but did not discuss the differences or conduct comparative experiments.
    - In LIO, players are symmetric and do not need to wait for an optimization problem to converge at one level (in the spirit of online cross-validation). And the incentivization occurs during the training phase without changing the game’s structure. So, could similar methods be directly applied?

- Coin Game might not highlight the advantages of the contract. A more suitable scenario would be a task with division of labor, such as the cleanup task. There is already work on MARL contracts in this scenario, "Formal Contracts Mitigate Social Dilemmas in Multi-Agent Reinforcement Learning." And LIO also uses this scenario.

Minors
- There is a large margin below Figure 1 and Figure 2.
- Some single quotation marks in the text should be changed to double quotation marks.

**Questions:**

1. How do the distinct and possibly conflicting objectives affect the meta-algorithm bilevel optimization procedure compared to same downstream tasks?
2. Nudging is an interesting solution for learning approximation error. However, who provides such nudging, i.e. in equations (20)-(22)? How does nudging change the discontinuous utility of the principle?
3. Besides, in the contracts of our general life, breach of contract is also an effective method to steer an agent’s behavior. What is the relationship between breach and nudging?

---

> ### Author Response · Authors · 2024-11-27
>
> Thanks for the careful review and comments.
>
> Thanks for the suggestion to try other more collaborative MARL tasks (such as division of labor tasks - e.g. clean up): We focused on tasks with conflicting objectives (such as in the coin game) as we believe these are the most challenging ones and hence most appealing ones for a contract approach (it’s not just about coordination of actions towards a joint objective, but intrinsically conflicting interest of the individual agents). It is also where we see the biggest gap in the literature. The rebuttal phase is too short to add further experiments, but we will be happy to work on adding other collaborative tasks to the list of experiments.
>
> Regarding technical depth, we would like to emphasize Theorem 3.4, which shows that the meta-algorithm is a contraction operator. This is crucial for the practical implementation using deep Q-learning, where the algorithms will need to be stopped short of convergence. We do validate the convergence/approximation properties empirically.
>
> **Re Q1:** We are not fully sure if we understand the question. Consider for a moment a single-shot contracting problem, the same rationale applies to the Markov Chain style games we consider. In the one shot game, a contract defines payments, and thus changes what the agent (or agents) consider as best responses to the contract. Intuitively, payments need to be higher if the agents’ interests are less well-aligned with each other and the principal. Conflicting interests jeopardize the stability of Q-learning, leading to instabilities or oscillations during the learning process. This has motivated us to propose and demonstrate that the meta-algorithm is a contraction, thereby enabling its practical implementation.
>
> **Re Q2:** Thanks! The idea here is that at equilibrium agents will necessarily be indifferent, making solutions prone to small errors in approximation. Think of the principal having an action plan in mind, with full knowledge, in an optimal contract agent(s) will be indifferent at each step where they have to make decisions. The idea behind nudging is that the principal can just increase payments a bit to robustify this solution. (In some sense move away from the discontinuous cliffs, into the interior of the best response regions.)
>
> **Re Q3:** Breach of contract. This is an interesting direction. But, of course, in many situations this would be prevented by legal provisions. We think that this could be an interesting future work to explore, but seems a bit like jumping two steps.

---

### Official Review · Reviewer_3zbp · 2024-11-03

**Soundness:** 3
**Presentation:** 3
**Contribution:** 2
**Rating:** 5
**Confidence:** 3

**Summary:**

This paper studies a problem where a principal must guide the actions of an agent (or several agents) to maximize the principal's utility using payments. This problem is formalized using a Markov decision process, where the principal chooses a policy that determines the payment to the agent depending on the state and outcome. The authors propose and analyze a simple meta-algorithm that ensures convergence to subgame perfect equilibrium (SPE). Moreover, experiments are conducted on the Coin Game environment, which is a medium-sized two-player social dilemma. The experiments show the effectiveness of the proposed approach for maximizing the principal's utility (in this case, social welfare).

**Strengths:**

- The intersection of reinforcement learning and contract design is an interesting area of study.
- The problem formulation is neat and well motivated.
- The paper is well-written and overall a pleasure to read.

**Weaknesses:**

- I struggle with identifying the technical novelty and the significance of the contributions:
	- From a technical standpoint, it appears that by considering SPE the problem becomes fairly simple and boils down to performing backwards induction (using standard RL techniques). In particular, as far as I can tell, the results of Theorem 3.3 and 3.4 follow straightforwardly from SPE+ well-known RL results. Could you highlight any specific novel technical challenges under the proposed problem setup?
	- Conceptually, the idea of a principal orchestrating agents in social dilemma to resolve miscoordination and maximize social welfare has been considered before (this is also essentially what much of mechanism design is about). In Appendix A.3, you argue that your work differs in perspective from prior work on "solving" social dilemmas by modifying the agents' rewards. In particular, to distinguish your work from the others you write "While the principal effectively modifies agents’ reward functions, the payments are costly." Are not the payments also costly in the models studied in prior work? For instance, in Yang et al. (2020), the agents transfer the rewards to other agents which is of course costly to the paying agent. Maybe the work of [Jackson and Wilkie (2002)](https://www.jstor.org/stable/3700662) is also relevant / of interest here.
- From my undestanding, principal-agent theory is broader than just contract design, and it would be helpful to the reader to discuss this (i.e., principal-agent theory) related work as well. For instance, [Myerson (1982)](https://www.sciencedirect.com/science/article/abs/pii/0304406882900064), [Zhang and Conitzer (2021)](https://arxiv.org/abs/2105.06008), [Gan et al. (2022)](https://arxiv.org/abs/2209.01146). There is also work on Bayesian persuasion which is part of the principal-agent problem family. Appendix A.2 does make it sound like principal-agent problems are just contract design problems.

**Questions:**

- $E_t$ in Proposition 4.1 is not defined in the main text. Overall, Proposition 4.1 is very hand-wavy. For example, you write that "This utility can be achieved through small(?) additional payments that counteract the agent's deviations, ...". I'm aware that more details are provided in the appendix, but the proposition should be stated rigorously or, alternatively, be made an informal remark.
- In your opinion, why would the centralized orchestrating of agents (i.e., the principal-agent perspective) be more appealing to resolve incentive issues than the large-scale decentralized individual bargaining perspective many other works take? (You mentioned some of these works in Appendix A.3).

---

> ### Author Response · Authors · 2024-11-27
>
> We thank the reviewer for the careful review and comments.
>
> Regarding technical novelty and significance: We view our work of theory-backed practical algorithm design. We provide theoretical evidence (e.g., Theorem 3.4, showing that the meta-algorithm is iteratively applying a contraction operator) that corroborates our ultimate approach, which consists in adopting standard (scalable) ML pipelines (deep Q-learning) to solve a challenging highly combinatorial optimization problem.
>
> It is true that prior work has considered similar approaches that modify agents’ rewards, but to the best of our knowledge the connection to contract design hasn’t been spelled out in this clarity. Contract design opens up new opportunities to design theoretically supported algorithms with the goal of minimizing the total payment. We believe that there is great value in bringing the two communities together.
>
> It is also true that principal-agent theory can be understood more broadly, and encompasses additional directions. We will be happy to emphasize this, and include additional discussion.
>
> Q1) Thanks for pointing this out. We will make sure to give this part another pass, and go with one of the two options that you suggest.
>
> Q2) This is an interesting question. We believe both approaches have their pros and cons, and are worth exploring. We believe that the more centralized approach that we pursue in this work could have a lot of value, thinking of platforms that will serve to aggregate services offered by AI agents.

---

> > ### Comment · Reviewer_3zbp · 2024-11-28
> >
> > Thank you for taking the time to respond.
> >
> > I agree that emphasizing the connection to contract design is interesting and you do this well. However, I still believe that the paper offers only very few novel technical / methodological contributions and insights beyond establishing a connection to contract design. I will keep my original borderline score.

---

### Official Review · Reviewer_qrta · 2024-11-06

**Soundness:** 3
**Presentation:** 3
**Contribution:** 3
**Rating:** 6
**Confidence:** 3

**Summary:**

The paper proposes to apply principal--agent theory to multi-agent RL with the goal of applying their methods to resolve social dilemma-like situations between AI agents, allowing different AI agents to collaborate more effectively. They define principal--agent MDP as a stochastic game with turns, where at each time-step principal observes the state and chooses its action (a contract), the agent observes state and principal's action and then chooses its action. As a solution concept, they choose the subgame-perfect equilibrium. They present a meta-algorithm that is essentially a bilevel optimization problem, which when solved, proven to find the SPE in their setting. Then they propose a Q-learning based approach to instead learn the SPE rather than computing it exactly. They present an early theoretical result connecting the approximation error in Q functions to a bound on principal's utility. They test their approach in the Coin Game environment in terms of both social welfare and convergence.

**Strengths:**

- The problem of orchestrating an ecosystem of AI agents to collaborate with each other is an interesting and relevant problem.
- The application of principal--agent theory and its combination with RL appears to be novel and also interesting.
- The paper is easy to read, and ideas are expressed clearly.

**Weaknesses:**

- The presented empirical and theoretical results appear too weak. Convergence of the meta-algorithm is not surprising, what is left out is the computational complexity of solving the bilevel optimization problem. Empirical results are also limited, and there is no comparison to other MARL algorithms that tackle social dilemmas (see the question).

- The connection to the motivating example seems lost once the Introduction is over. More discussion at the end about how this allows for orchestrating of AI agents is needed.

**Questions:**

1. Is there a reason why you have not benchmarked your method against other MARL methods that allow agents to gift each other rewards? For example [1,2,3]? I would expect these to be at least discussed as an alternative approach.

2. For the case of orchestrating AI agents (your motivating example), what would be the advantage of your principal--agent framework over things like adaptive mechanism design?

[1] Yang, Jiachen, et al. "Learning to incentivize other learning agents." Advances in Neural Information Processing Systems 33 (2020): 15208-15219.

[2] Lupu, Andrei, and Doina Precup. "Gifting in multi-agent reinforcement learning." Proceedings of the 19th International Conference on autonomous agents and multiagent systems. 2020.

[3] Kolumbus, Yoav, Joe Halpern, and Éva Tardos. "Paying to Do Better: Games with Payments between Learning Agents." arXiv preprint arXiv:2405.20880 (2024).

---

> ### Author Response · Authors · 2024-11-27
>
> Thanks for the careful review and comments.
>
> We believe that our work has some non-trivial theoretical insights, such as Theorem 3.4, which shows that the meta-algorithm is applying a contraction operator. This is crucial for the applicability of our results, because in the experimental parts we need to stop the deep Q-learning implementation short of convergence.
>
> It is also not true that we don’t compare to existing baselines in the MARL literature (see reply to Q1 below). We do apologize that this was a bit hidden (we should have explicitly referred to the paper that inspired us to adopt this benchmark for comparison).
>
> **Re Q1:** Thanks for pointing us to [2,3] (we already cite [1]), we will be happy to include additional discussion. For gifting methods as baselines, we would like to stress that these are not directly comparable with our method as agents are subtracted rewards and there is no centralized principal; we already compare with this line of work by adapting a similar method of the other contract design in MARL paper (see citation below). This is the “constant baseline” we refer to in Section 5 and Figures 2a and 2b. We found that this alternative is much less effective, achieving only around 85-90% of the welfare that our approach achieves (see the gap between the orange and gray lines in Figure 2a).
>
> Phillip JK Christoffersen, Andreas A Haupt, and Dylan Hadfield-Menell. Get it in writing: Formal contracts mitigate social dilemmas in multi-agent RL. In Proceedings of the 2023 International Conference on Autonomous Agents and Multiagent Systems, pp. 448–456, 2023.
>
> **Re Q2:** There is a profound difference between an “adaptive contract design approach” (our work) and an “adaptive mechanism design approach”. In short: These are two different branches in economics, dealing with very different incentive constraints. The emphasis on the former is on incentivizing effort, and a main obstacle is that the principal cannot directly observe how much effort the agents exert. We believe that this “incentivize effort” perspective is very adequate when thinking about creating marketplaces for AI agents, as these are quintessentially markets for services (effort).
>
> This notwithstanding, we believe that ultimately we may want a solution that marries the two directions. Indeed, the focus of the adaptive mechanism design approach is that agents may have hidden types (think capabilities), and we would like to design mechanisms that incentivize agents to truthfully report their private types.
>
> We would like to point to two simultaneous/subsequent works that also argue in favor of a contract design approach for AI agents. We believe that this provides additional evidence that this is a timely and important perspective.
>
> Jibang Wu, Siyu Chen, Mengdi Wang, Huazheng Wang, Haifeng Xu. Contractual Reinforcement Learning: Pulling Arms with Invisible Hands. In ArXiv, 2024. https://arxiv.org/abs/2407.01458
>
> Matteo Bollini, Francesco Bacchiocchi, Matteo Castiglioni, Alberto Marchesi, Nicola Gatti. Contracting with a Reinforcement Learning Agent by Playing Trick or Treat. In ArXiv, 2024. https://arxiv.org/abs/2410.13520

---

> > ### Comment · Reviewer_qrta · 2024-12-02
> >
> > Thank you for your response. This clarifies the baseline question for me. To be clear, in gifting mechanisms, there are many different models such as: giving each agent a separate gift budget or making gift sending a no-cost event. These could have been tested against your approach. Also, even if sending gifts would cost to the sender, this still does not invalidate them as a baseline approach to orchestration. Reward is a made up quantity in reinforcement learning, and unlike economics and game theory, does not need to correspond to things like monetary value or actual utility gain. It is a signal that induces desired behaviour.
> >
> > Similarly, your comparison to adaptive mechanism design is also not clear to me. Adaptive mechanism design is literally about learning how to influence the payoff structure in order to induce desired equilibrium behaviour amongst players. For work on this that is very similar to your objective see:
> >
> > "We consider the problem of how an
> > external agent can promote cooperation between artificial learners by
> > distributing additional rewards and punishments based on observing
> > the learners’ actions. We propose a rule for automatically learning
> > how to create the right incentives by considering the players’ anticipated parameter updates." [1]
> >
> > [1] Baumann, Tobias, Thore Graepel, and John Shawe-Taylor. "Adaptive mechanism design: Learning to promote cooperation." 2020 International Joint Conference on Neural Networks (IJCNN). IEEE, 2020.
> >
> >
> > ** Overall: ** I have a positive sense about the paper, so I will maintain my score. But I believe someone other than me must champion its acceptance.

---

### Meta-Review · Area_Chair_5FLh · 2024-12-22

**Metareview:**

This paper reduces mechanism design for coordinating AI agents to solving a turn-based stochastic game. I found the paper to be well-written and easy to follow. However, both the reviewers and I share the concern regarding the limited technical novelty and the significance of the contribution.

The reduction from contract design to stochastic games is fairly straightforward, and the fact that turn-based stochastic games can be solved using simple DP is largely considered folklore within the community. Additionally, Reviewer 3zbp noted that the conceptual idea of a principal orchestrating agents in social dilemmas to address miscoordination and maximize social welfare has been explored in prior work.

As a result, I feel that the contribution of this paper is marginal and falls below the standard for acceptance.

**Additional Comments On Reviewer Discussion:**

The primary concerns raised by the reviewers revolve around the novelty and technical depth of the work. I feel these issues were not adequately addressed during the rebuttal phase, which is the main reason I lean toward rejection.

---

### Decision · Program_Chairs · 2025-01-22

Reject